

# Temperature variability of the Iberian Range since 1602 inferred from tree-ring records

**E. Tejedor[1,2,3], M.A. Saz[1,2], J.M. Cuadrat[1,2], J. Esper[3], M. de Luis[1,2]**

[1]{University of Zaragoza, 50009 Zaragoza, Spain}

[2]{Environmental Sciences Institute of the University of Zaragoza }

[3]{Department of Geography, Johannes Gutenberg University, 55099 Mainz, Germany}

Correspondence to: E. Tejedor (etejedor@unizar.com)

**Abstract**

Tree-rings are an important proxy to understand the natural drivers of climate variability in the Mediterranean basin and hence to improve future climate scenarios in a vulnerable region. Here, we compile 316 tree-ring width series from 11 conifer sites in the western Iberian Range. We apply a new standardization method based on the trunk basal area instead of the tree cambial age to develop a regional chronology which preserves high to low frequency variability. A new reconstruction for the 1602-2012 period correlates at -0.78 with observational September temperatures with a cumulative mean of the 21 previous months over the 1945-2012 calibration period. The new $IR2T_{max}$ reconstruction is spatially representative for the Iberian Peninsula and captures the full range of past Iberian Range temperature variability. Reconstructed long-term temperature variations match reasonably well with solar irradiance changes since warm and cold phases correspond with high and low solar activity, respectively. In addition, some annual temperatures downturns coincide with volcanic eruptions with a three year lag.

## 1 Introduction

The IPCC report (IPCC, 2013) highlighted a likely increase of average global temperatures in upcoming decades, and pointed particularly to the Mediterranean basin, and therefore in the



Iberian Peninsula (IP), as a region of substantial modelled temperature changes. The
Mediterranean area is located in the transitional zone between tropical and extra-tropical
climate systems, characterized by a complex topography and high climatic variability (Hertig
and Jacobeit 2008). Taking into account these features, even relatively minor modifications of
the general circulation, i.e. a shift in the location of sub-tropical high pressure cells, can lead
to substantial changes in Mediterranean climate (Giorgi and Lionello 2008), making the study
area a potentially vulnerable region to anthropogenic climatic changes by anthropogenic
forces, i.e. increasing concentrations of greenhouse gases (Lionello et al., 2006a; Ulbrich et
al., 2006)
Major recent efforts have been made in understanding trends in temperatures throughout the
IP over the instrumental period (Kenaway et al., 2012; Pena-Angulo et al., 2015; Gonzalez-
Hidalgo et al., 2015) and future climate change scenarios (Sánchez et al., 2004; López-
Moreno et al., 2014). However, the fact that most of the observational records do not begin
until the 1950s (Gonzalez-Hidalgo et al., 2011) is limiting the possibility of investigating the
inter-annual to multi-centennial long-term temperature variability. Therefore, it is crucial to
explore climate proxy data and develop long-term reconstructions of regional temperature
variability to evaluate spatial patterns of climatic change and the role of natural and
anthropogenic forcings on climate variations (Büntgen et al., 2005). In the IP, much progress
has been made to reconstruct past centuries climate variability, including analysis of
documentary evidences for temperature (i.e. Camuffo et al., 2010) and droughts
reconstruction (i.e. Barriendos et al. 1997; Cuadrat and Vicente, 2007; Domínguez-Castro et
al., 2010). Additionally, progress has been made to further understanding of long-term climate
variability of the IP through dendroclimatological studies focussing on drought (Esper et al.,
2014; Tejedor et al., 2015) and temperature (Büntgen et al., 2008; Dorado-Liñán et al., 2012,
2014; Esper et al. 2015a). Nevertheless, a high-resolution temperature reconstruction for
central Spain is still missing.
Several studies have been made to develop a temperature reconstruction for the Iberian Range
(IR) using *Pinus uncinata* tree-ring data (Creus and Puigdefabreas, 1982; Ruiz, 1989). The
results, in fact, showed a pronounced inter-annual to century scale chronology variability.
However, their main result was a complex growth response function due to a mixed climate
signal instead of a temperature reconstruction. Furthermore, Saz (2003) developed a 500-year
temperature reconstruction for the Ebro Depression (North of Spain), but this chronology is





based on a reduced number of cores and a standardized methodology that did not retain the
medium and low frequency variance.
Here we present the first tree-ring dataset combining samples from three different sources
from the eastern IR extending back from the Little Ice Age (1465) to present (2012). The aim
of this study is to develop a temperature reconstruction representing the IR, and thereby fill
the gap between records located in the northern and southern IP. A new methodology, based
on basal area instead of the cambial-age, was applied to preserve high-to-low frequency
variance in the resulting chronologies. Furthermore, the relationship between the tree-ring and
climate data is reanalysed by adding memory to the climate parameters, since memory effects
on tree-ring data are much less acknowledged (Anchukaitis et al., 2012). This analysis is
challenging because of the mix of tree species and their unidentified responses to climate. The
resulting reconstruction of September maximum temperatures over the past four centuries is
compared with latest findings from the Pyrenees and Cazorla, and the relationship with solar
and volcanic forcings at inter-annual to multi-decadal timescales.
**2   Material and methods**
**2.1   Site description**
We compiled a tree ring network from 11 different sites in the western IR (Table 1) in the
province of Soria. Urbión is the most extensive forest of the IP including 120,000 ha between
the Burgos and Soria provinces. It has a long forest management tradition. Therefore, all sites
are situated at high elevation locations where forests are least exploited and maximum tree
age is reached (Fig.1). The altitude of the sampling sites ranges from 1,500 to 1,900 meters
above sea level (masl) with a mean of 1,758 masl. These forests belong to the Continental
Bioclimatic Belt (Guijarro, 2013) characterized by moderate mean temperatures (9.5°C,
Fig.2B) and a large seasonal range including more than 90 frost days and summer heat
exceeding 30ºC . Mean annual precipitation for the period 1944-2014 is 927 mm (CRU TS.3
v.23 dataset by Harris et al., 2014) and reaches its maximum during December (Fig. 2AC).
Although scotts pine (*Pinus sylvestris*) is the dominant tree species of the region, other
pinaceaes are found such as *Pinus pinaster*, *Pinus nigra* or *Pinus uncinata*. Especially
remarkable is occurrence of *Pinus uncinata* growing above 1,900 masl and reaching its



European southern distribution limits in the IR. The lithology of the study area consists of
sandstones, conglomerates and lutites.
**2.2   Tree ring chronology development**
The new dataset is composed by 316 tree-ring width (TRW) series of *Pinus uncinata* (56) and
*Pinus sylvestris* (260) located in the western IR (Tab. 1, Fig. 1). The most recent samples
were collected during the field campaign in 2013 including old dominant and co-dominant
trees with healthy trunks and no sign of human interference. We extracted two core samples
from each tree at breast height (1.3 m) when possible, otherwise, we try to avoid compression
wood due to steep slopes, compiling a set of 96 new samples from two sites, i.e. the outermost
ring is 2012. Core samples were air-dried and glued onto wooden holders and subsequently
sanded to ease growth ring identification (Stokes and Smiley 1968). The samples were then
scanned  and  synchronized  using  CoRecorder  software  (Larsson  2012)  (Cybis
Dendrochronology 2014) to identify the position and exact dating of each ring. The tree-ring
width was measured, at 0.01 mm precision, using LINTAB table (Rinn 2005). Prior to
detrending, COFECHA (Holmes 1983) was used to assess the cross-dating of all
measurement series.
An additional set of 95 samples from three sites was provided by the project CLI96-1862
(Creus et al. 1992, Saz 2003) i.e., the outermost rings range from 1992 to 1993. Finally, a set
of 125 samples from five sites was downloaded from the International Tree Ring Data Bank
(ITRDB,      http://www.ncdc.noaa.gov/data-access/paleoclimatology-data/datasets/tree-ring).
These data were developed in the 1980s by K. Richter and collaborators, i.e. the outermost
rings range from 1977 to 1985.
In order to attempt a climate reconstruction for the western IR from this tree-ring network, we
perform an exploratory analysis of the 11 tree-ring chronologies by creating a correlation
matrix of the raw chronologies for each site for the common period (1842-1977) and for the
full period (1465-2012).
2.2.1  Standardization methods
The key concept in dendroclimatology is referred to as the standardization process (Fritts,
1976; Cook et al., 1990) where the aim is to preserve as much of the climate-related
information as possible while removing the non-climatic information from the raw TRW





measurements. However, with most of the standardization methods a varying proportion of
the low-frequency climatic information is also lost in the process (Grudd, 2008). When the
aim is to use tree-ring chronologies as a proxy for climatic reconstructions, an adequate
standardization is critical and the best method should preserve high to low frequency
variations (Büntgen et al., 2004). It is common practice to calculate a mean value function as
the best estimate of the trees' signal at a site (Frank et al., 2006).
We here applied four standardization methods to the 316 TRW measurement series to develop
a single tree-ring index chronology. (i) To emphasize inter-decadal and higher frequency
variations, each ring width series was fitted with a cubic spline with a 50% frequency
response cut off at 67% of the series length (Cook et al., 1990). A bi-weight robust mean was
calculated to assemble the ArstanSTD regional chronology. (ii) A residual chronology
(ArstanRES) is produced after removing first-order autoregression to emphasize high-
frequency variability. (iii) To preserve common inter-decadal and lower frequency variations,
Regional Curve Standardization (RCS) was applied (Mitchell, 1967; Briffa et al., 1992, 1996;
Esper et al., 2003). RCS is an age-dependent composite method and involves dividing the size
of each tree-ring by the value expected from its cambial age. To assemble the chronology, all
the series are aligned by cambial age. A single growth function (regional curve, RC)
smoothed using a spline function of 10% of the series length is fit to the mean of all age-
aligned series. A biweigth robust mean was applied to develop the RCS chronology (RCS).
(iv)To preserve high to low frequency variance, we additionally applied a novel
standardization method based on the principles of RCS. However, instead of using the
cambial age of the trees as the independent variable, we used their sizes, calculated as the
square of the basal area of the tree in the year prior to ring formation. Then, a Poisson
regression model was used to fit the individual tree-ring widths. Standardized indices were
calculated as the ratio between the observed and predicted values, and a biweigth robust mean
was used to develop the Basal Area Poisson chronology (BasPois).
To evaluate uncertainty of the mean chronologies running interseries correlations (Rbar) and
the express population signal (EPS) were calculated (Wigley et al., 1984). Rbar is a measure
of the strength of the common growth 'signal' within the chronology (Wigley et al. 1984;
Briffa and Jones, 1990), here calculated in a 50-year window sliding along the chronology.
EPS is an estimate of the chronology's ability to represent the signal strength of a chronology
on a theoretical infinite population (Wigley et al., 1984).





**2.3 Climatic data, calibration and climate reconstruction**
Monthly temperature (mean, maximum, and minimum) and precipitation values from the
gridded CRU TS v.3.22 dataset (0.5º resolution) dataset for the period 1945-2012 were used
(Harris et al. 2014). The three grid points closest to the tree-ring network were averaged to
develop a regional time series (Fig. 1). In addition, we calculate a cumulative monthly mean
for each of the four parameters (max., min., mean temperature, and monthly precipitation).
The cumulative mean is calculated by adding the months gradually. First the previous month
is added, and then further months are included up to 36 previous months. For the calculations
we take into account the current and the previous year.
For calibration, we correlated the four chronologies (ArstanSTD, ArstanRES, RCS, and
BasPois) with monthly climate data and the cumulative monthly mean derived. To assess the
stability of the correlation, we calculated a 30-year moving correlation shifted along 1945-
2012 with the cumulative monthly mean from the current and the previous year. In addition,
the maximum and minimum differences between the moving correlations were calculated. As
a result, the climatic variable chosen for the reconstruction is supported by having the highest
moving correlation with the least difference between the maximum and the minimum over the
moving correlation period.
A split calibration/verification approach was perform over the periods 1945-1978 and 1979-
2012 to evaluate the accuracy of the transfer model considering the following metrics;
Pearson's correlation (r), coefficient of determination ($r^2$), reduction of error (RE), mean
square error (MSE), and sign test (Cook et al., 1994). R is a measure of the linear correlation
between the chronology and climatic variable. $R^2$ indicates how well the data fit a statistical
model. An $r^2$ of 1 indicates that the regression line perfectly fits the data; an $r^2$ of 0 indicates
that there is not fit at all. RE is a measure of shared variance between actual and estimated
series and provides sensitive measure of the reliability of a reconstruction (Cook et al., 1994;
Akkemik et al., 2005; Büntgen et al., 2008); it ranges from +1 indicating perfect agreement, to
minus infinity. MSE estimates the difference between the modelled and measured while sign
test compares the number of agreeing and disagreeing interval trends, from year-to-year,
between the observed and reconstructed series (Fritts et al., 1990; Cufar et al., 2008).
Additionally, a Superposed Epoch Analysis (SEA; Panofsky and Brier, 1958) was performed
using dplR (Bunn, 2008) to assess post-volcanic cooling signals in our reconstruction. The
approach has been used in studies of volcanic effect on climate (Fischer et al., 2007; D'Arrigo



et al., 2009; Esper et al. 2013a, 2013b). The major volcanic events chosen for the analysis
were those identified by Crowley (2000).
To transfer the TRW chronology into a temperature reconstruction a linear regression model
was used. The magnitude and the spatial extent of the climate signal are evaluated considering
the CRU TS v. 3.22 gridded dataset for Europe.
**3    Results**
The correlation matrix (Fig. 3) shows the high inter-correlation between sampling sites and
tree species. The highest correlation is found between *Pinus uncinata* (VIN and CAV) located
at the highest altitude. On the other hand, the weakest correlation is found between one of the
lowest sites (s006) and the highest (VIN). The mean correlation among all sampling sites is r
= 0.51 over the common period (1842-1977) is 0.51, and r = 0.46 over the full period of
overlap, revealing a regionally common, external forcing controlling tree growth and
justifying the development of a single chronology integrating the data from this IP tree-ring
network.
The model (regional curve) of the RCS standardization method and the model of the BasPois
method are presented in Fig.4. BasPois model (Fig.4a) indicates a growth of 130 mm when
the size of the basal area is near 0 and a growth of 8mm when it reaches the maximum basal
area. RCS model (Fig.4b) presents values of 250 mm of growth when the cambial age is 0
with a gradual decline of the growth until the cambial of 450. Cambial age from 500 to 550
has a slight increase in growth most likely derived by low replication regarding trees with this
age.
Calibration of the four differently detrended mean chronologies reveals a highly negative
correlation with maximum temperatures (Fig. 5). The ArstanRES chronology shows moderate
correlations with in previous-year September (r = -0.25), and the ArstanSTD chronology
correlates at r = -0.38 with June and September temperature of the previous year. Considering
the RCS chronology, the previous-year September signal increases to r = -0.49with a
cumulative monthly mean of 21 months. Finally, the best correlations is revealed for the
BasPois chronology reaching r = -0.78 with maximum September temperature of the previous
year with a cumulative mean of 21 months, which is, in fact a two year cumulative monthly
mean. Even though the signals show the same seasonal patterns among the chronologies, the





BasPois record always shows the highest correlations. Accordingly, we used the BasPois
chronology for the calibration and reconstruction process.
The final BasPois network chronology (Fig.6) is based on 316 TRW series of *Pinus uncinata*
and *Pinus sylvestris* spanning the 1465-2012 period. Since this chronology is derived from
only living trees, mean chronology age increases from 47 years in 1966 to 528 in 1465. The
mean sensitivity is 0.21, and first-order autocorrelation 0. The inter-series correlation (Rbar)
reaches 0.26, and the first principal component explains about 35% of the variance. The
network chronology's signal to noise ratio is 48.52, and EPS exceeds 0.85 after 1602,
constraining the reconstruction period to 410 years until 2012.
The selection of the best climate parameter to develop the reconstruction is presented in the
Figure 7. Correlations between -0.54 and -0.86 representing only the most significant values
are shown. Four parameters reveal the highest correlations over the full calibration period:
October of the current year with a cumulative monthly mean of 22 months; September of the
previous year with a cumulative monthly mean of 20-months; September of the previous year
with a cumulative monthly mean of 21months; and October of the previous year with a
cumulative monthly mean of 21 months. The stability of the correlation and therefore the
consistency of the signal are tested considering the minimum difference between the
maximum and minimum correlation (Fig. 7b) over the full running correlation period. The
smallest difference (0.24) is reached for September of the previous year with a cumulative
monthly mean of 21months. Therefore, this parameter is chosen for the climate
reconstruction. According to the 30-year moving correlations, maximum values are reached
from 1973-2003 (r = -0.80), whereas the lowest 30-year correlation (r = -0.60) is reached from
1956-1986. In addition, the relationship between September of the previous year with a
cumulative monthly mean of 21months is spatially consistent throughout the Iberian
Peninsula, reaching into southern France and northern Africa (Fig.11).
The transfer model is validated by the high correlation (r = -0.78) and significant correlation
coefficients ($r^2$ = 0.61) over the full period 1945-2012. Through the split
calibration/verification process, considering 1945-1978 and 1979-2012, the temporal
robustness was tested revealing highly significant correlations for both periods ($r^2$=0.41 and
$r^2$=0.55 respectively) and verifying the final reconstruction (Table 2 and Fig. 8). To develop
the final reconstruction spanning 1602-2012, we used a lineal regression model over the full





period 1945-2012 with maximum temperature of September of the previous year with a
cumulative monthly mean of 21months (Eq.1), denominated IR2T$_{max}$:
$IR2T_{max} = -01533 * BasPoisChron + 2.3542 (r^2{}_{adj} = 0.61; p < 0.0001)$.      (1)
### 3.1  IR2T$_{max}$ reconstruction
IR2T$_{max}$ describes 410 years of maximum temperature of September with a cumulative
monthly mean of 21-months meaning it has memory of the last two years. Temperature
ranges from 13.52ºC (-2.13ºC with respect to the mean) in 1603 to 17.64ºC n (+1.94ºC with
respect to the mean) in 2005 (Fig. 9). It is remarkable that the 12 years of the XXI century
happen to be within the 25 warmest years. IR2T$_{max}$ covers a part of the Little Ice Age (Grove,
1988) from 1602 to the end of the XIX century. The year-to-year temperature variability is
3.92ºC in the seventeenth century, 2.89ºC in the eighteen century, 3.17ºC in the nineteenth
century and 3.07ºC in the twentieth century. The seventeenth and eighteen centuries were the
coldest of the reconstruction with 73% and 80% of the years with temperatures below the
long-term mean, respectively. On the other hand, the nineteenth and the twentieth centuries
were the warmest with 66% and 78% of the years exceeding the mean.
The main driver of the large-scale character of the warm and cold episodes may be changes in
the solar activity (Fig.9). The beginning of the reconstruction starts with the end of the Spörer
Minimum. The Maunder minimum, from 1645 to 1715 (Luterbach et al., 2001) seems to
cohere with a cold period from 1645 to 1706. In addition, the Dalton minimum from 1796 to
1830, is detected for the period 1810 to 1838. However, a considerably cold period from 1778
to 1798 is not in consonance with a decrease in the solar activity. Four warm periods, 1626-
1637, 1800-1809, 1845-1859 and 1986-2012, have been identified to cohere with increased
solar activity. Overall, the correlation between the reconstruction and the solar activity is 0.34
($p < 0.0001$), and increases to r = 0.49 after 11-year low pass filtering the series, thought the
degrees of freedom are substantially reduced due to the increase autocorrelation.
The SEA (Fig.10) indicates some impact of volcanic eruptions on the short-term temperature
variability within the reconstruction. It shows significance ($p < 0.05$) decrease in September's
temperature with a lag of three years.
Figure 11 shows the spatial correlation between the reconstruction and the CRU TS v.3.22 for
Europe and northern Africa. High adjusted correlations ($r^2$>0.4, $p < 0.0001$) indicate a robust



agreement and spatial extend of the reconstruction over the Iberian Peninsula (IP), especially
for the central and Mediterranean Spain. The spatial correlation, however, decreases towards
the southwest of the IP and the north of Europe.
**4   Discussion and conclusion**
Based on a coherent network of 11 tree-ring sites in the IR including 316 TRW series we
developed a 410-year maximum September temperature reconstruction. This record is the first
climate reconstruction for the IR filling the gap between the temperature reconstructions
developed for the north IP (Büntgen et al., 2008; Dorado-Liñán et al., 2012a, Esper et al.
2015a) and for the southern IP (Dorado-Liñán et al, 2014). The IR2T$_{max}$ has been achieved
using TRW as well as for the southern IP (Dorado-Liñán et al, 2014). However, for the
Pyrenees, MXD (Büntgen et al., 2008, Dorado-Liñán et al., 2012a) or stable isotopes (Esper et
al. 2015a) are needed to get skilful records for a temperature reconstruction.
The main statistics used to verify the accuracy of the reconstruction present similar values to
those developed for the IP. For instance, the best RE coefficient is 0.99 for the split
calibration/verification modelled meaning that the reconstruction has almost the perfect skill.
A relatively high signal to noise ratio indicates there is meaningful climatic information in the
chronology. The mean correlation between sites for the common period (r = 0.51, Fig. 3)
reveals substantial agreement between the sites and species. Correlation is strongest among
high elevation sites including the sites VIN and CAV which are both derived from *Pinus*
*uncinata*. The mean chronology, with 35.40% of the first component variance and 48.52 of
signal to noise ratio, captures the regional climate signal accurately, which highlights the
beauty of regional averages (Briffa et al., 1998).
The original, raw chronology extended over the 1465-2012 period, some 150 years longer
than the final reconstruction. However, due to low EPS values prior to 1602, which is related
to the low number of samples the final reconstruction was developed for the period 1602-

27   2012.

A novel detrending approach, considering a Basal Area-Poisson model instead of the
traditional regional curve (Esper et al. 2003) has certainly improved the skill of the
reconstruction and enabled retaining high-to-low frequency climate variance. The traditional
approach of using RCS with the mean TRW curve of the age-aligned data only reached



correlations with the maximum temperature of September with a cumulative monthly mean of
21months up to r = -0.5, while with the new approach reached r = -0.78.
It is usually difficult to determine the extent to which the effects of environmental factors on
tree growth depend on age (genetic control) and/or on size (physiological control), but recent
investigations suggest that it is often the size, and not the age, that is important (Mencuccini et
al. 2005; Peñuelas 2005). In fact, climate variability is more size-dependent than age or
species (De Luis et al., 2009). Hence, the size-based standardization considered here
maximizes the common signal. In addition, when combining TRW series from different sites
and species, as done here, the heterogeneity in responses might be large. Therefore, size
standardization may be a commendable solution to develop unbiased chronologies. Finally,
the new method should be tested in other locations since it may help to maximizes responses
especially in heterogeneous areas.
Taking into account that TRW growth is conditioned by the storage of starch and sugar in
parenchyma ray tissue, the remobilization of carbohydrates from root structures, and the
development of needle enduring several growing seasons, influencing the radial increment
beyond the instant impact of temperature variability (Pallardy, 2010), we added the
cumulative monthly mean to the climate parameters. In fact, we demonstrated that the signal
is magnified with a memory of 21 months from the previous September. Thus, developing the
two year memory $IR2T_{max}$ allowed us to maintain not only the low frequency signal,
highlighting the warm and cold phases, which may be explained by the high correlation with
solar activity during 410 years (0.34, $p<0.001$), but also the high frequency signal,
emphasizing the memory effects of the volcanic eruptions in TRW, already studied by Briffa
et al. (1998) and recently by Esper et al. (2015b). According to the SEA (Fig.9), the volcanic
eruptions have a significance reduction (95% confidence) of September's temperature (-
1.98ºC) with a three years lag. However, the $IR2T_{max}$ is already considering the two previous
year's temperature, which means the temperature decrease occurred the year after the extreme
volcanic event in consistency with (Frank et al., 2007a). The stability of the signal was
assessed by a 30-y moving correlation from 1945 to 2012, which shows a better correlation
for the period 1979-2012 in agreement with the raise of temperatures observed for last
decades which may be limiting TRW growth and therefore magnifying the climate signal.
However, the relationship between the chronology and the climate parameter chosen never
drops from -0.54 within the calibration period 1945-2012. The negative correlation with



maximum temperature of previous September is in concordance with the values detected in
Cazorla by Dorado-Liñán et al. 2014. Presumably, a continuous rise in temperatures, as
suggested by the IPCC (2013), will trigger an incessant decrease in the tree-ring growth.
Even though the CRU dataset extents the 1901-2013 period, the general distribution of
meteorological observatories in Spain did not begin until the mid-twentieth century
(Gonzalez-Hidalgo et al. 2011). In fact, the closest instrumental weather station, located in
Vinuesa (Fig.1), began in 1945. However, due to the large amount of gaps in the time series,
the CRU dataset was used instead for the split calibration/verification approach for the period
1945-2012. The advantages of regional climatic averages were already addressed by Blasing
et al. (1981) stating that the average climatic record of the gridded dataset over the study area
is representative of the regional climatic conditions, and does not reflect microclimate
conditions which may be characteristic of the climatic record at a single station. Tree-ring
data might therefore have more variance in common with the regionally averaged climatic
record than with the climatic record of the nearest weather station. Generally, studies have
shown that the measurements of MXD produce chronologies with an improved climatic signal
(Briffa et al., 2002) as it was revealed for summer temperature reconstructions (Hughes et al.,
1984; Büntgen et al. 2008; Matskosvsky and Helama, 2014). However, based on a TRW
chronology, it is remarkable the high correlation coefficient for the full calibration period and
the CRU dataset (r = -0.78).
Throughout the $IR2T_{max}$ reconstruction we identified the main warm and cold phases
(Maunder minimum, Dalton minimum) related with long-term temperature variability
generally attributed to changes in cycles of activity (Lean et al., 1995; Lassen et al. 1995;
Haigh et al. 2015). In addition, similar cold and warm phases are observed comparing with
the Pyrenees (Büntgen et al. 2008) and Cazorla (Dorado-Liñán et al. 2014) reconstructions.
However, previously to the Dalton minimum, a warm phase is detected in $IR2T_{max}$ and the
Cazorla reconstruction although it is not present in the Pyrenees or in the Alps (Büntgen et al.,

27   2011).

Through the spatial extent and magnitude of the $IR2T_{max}$ reconstruction over Europe it can be
acknowledged that the reconstruction is effective and usable for most of the Spanish Iberian
Peninsula. Working especially for the central and Mediterranean IP with very high
correlations ($r^2$>0.4).





## 1 Acknowledgements

This study was supported by the Spanish government (CGL2011-28255) and the government of Aragon throughout the Program of research groups (group Clima, Cambio Global y Sistemas Naturales, BOA 147 of 18-12-2002) and FEDER funds. Ernesto Tejedor is supported by the government of Aragon with a Ph.D. grant. Fieldwork was carried out in the province of Soria; we are most grateful to its authorities, for supporting the sampling campaign. We are thankful to Klemen Novak, Edurne Martinez, Luis Alberto Longares, and Roberto Serrano for help during fieldwork.



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





1    Table 1. Tree ring sites characteristics

| Code | Site | Source | Lat | Long | Elevation | Species | Tree no | Sample no | Tree-rings | Period |
|---|---|---|---|---|---|---|---|---|---|---|
| s047 | Urbión Covaleda | ITRDB | 41.98 | -2.87 | 1750 | PISY | 15 | 31 | 6549 | 1567-1983 |
| s048 | Urbión Duruelo | ITRDB | 42.02 | -2.90 | 1840 | PISY | 8 | 17 | 3590 | 1671-1983 |
| s049 | Urbión Quintenar | ITRDB | 42.03 | -3.03 | 1840 | PISY | 12 | 27 | 4713 | 1593-1985 |
| s050 | Urbión Vinuesa | ITRDB | 42.00 | -2.85 | 1750 | PISY | 4 | 8 | 1942 | 1681-1983 |
| s006 | Urbión | ITRDB | 42.03 | -2.7 | 1634 | PISY | 11 | 22 | 2397 | 1842-1977 |
| CAV | Castillo de Vinuesa | UNIZAR | 42.01 | -2.75 | 1900 | PIUN | 18 | 36 | 9236 | 1593-2012 |
| COV | Covaleda | IPE-CSIC-UNIZAR | 41.93 | -2.83 | 1500 | PISY | 16 | 48 | 14696 | 1568-1993 |
| HER | Barranco de las heridas | IPE-CSIC-UNIZAR | 41.94 | -2.84 | 1500 | PISY | 25 | 32 | 9347 | 1562-1993 |
| NEI | Neila | IPE-CSIC-UNIZAR | 42.05 | -3.08 | 1850 | PISY | 9 | 15 | 4822 | 1587-1992 |





| | | | | | | | | | |
|---|---|---|---|---|---|---|---|---|---|
| URB | Picos de Urbión | UNIZAR | 41.96 | -2.82 | 1750 | PISY | 28 | 60 | 11328 | 1733-2012 |
| VIN | Castillo de Vinuesa | IPE-CSIC-UNIZAR | 42.03 | -2.73 | 1900 | PIUN | 13 | 20 | 7653 | 1465-1992 |
| | | | | Total | 159 | 316 | 76273 | |

*UNIZAR* University of Zaragoza, *IPE-CSIC* Spanish National Research Council, *ITRDB* International Tree-Ring
Databank

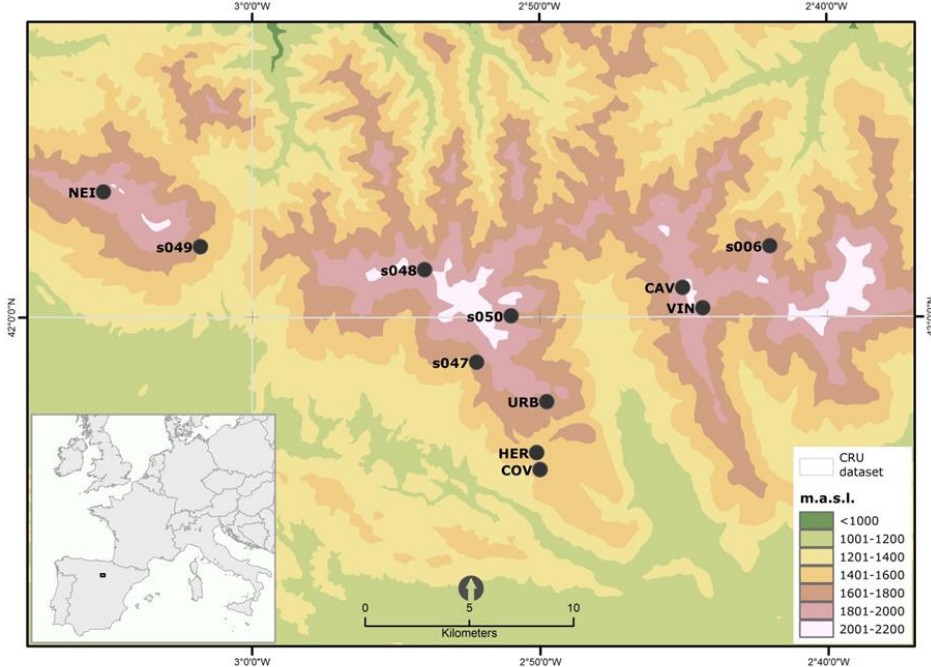

Figure 1. Map showing the tree ring study sites and the climate data (CRU TS v.3.22) grid
points in the Western Iberian Range (Soria).





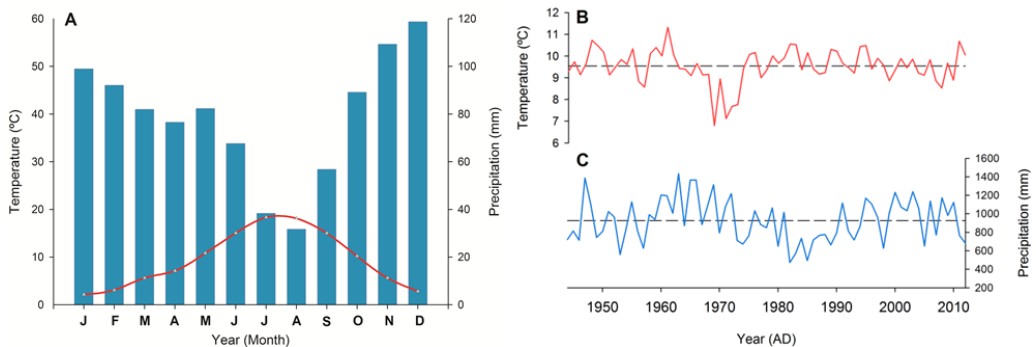

Figure 2. Climate diagram (A), mean temperature (B), mean precipitation (C) calculated using data from CRU TS v.3.22 over the period 1944-2012 (Harris et al 2014).

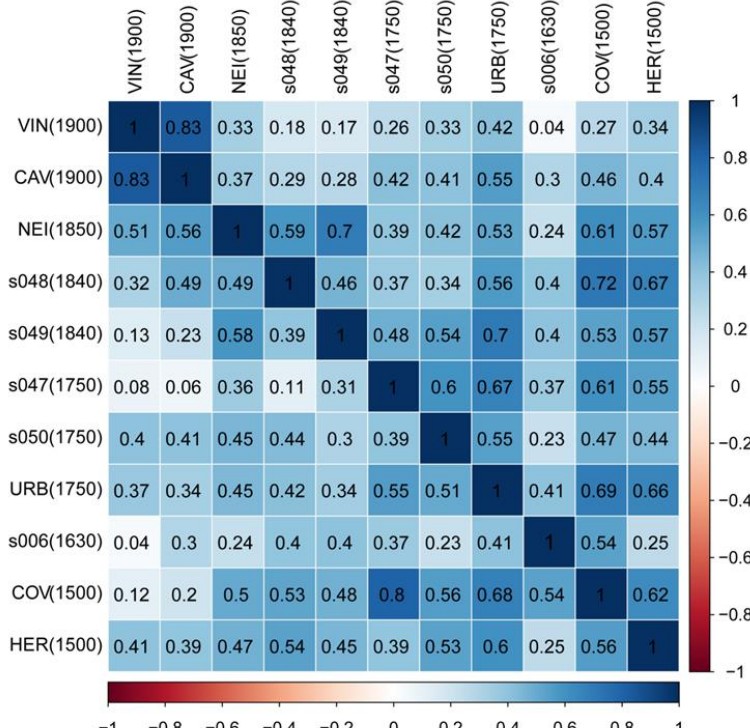

Figure 3. Correlation of the raw chronologies sorted by elevation. Top right shows the correlations calculated over the common period 1842-1977. Bottom left shows the correlation over the full period of overlap between pairs of chronologies



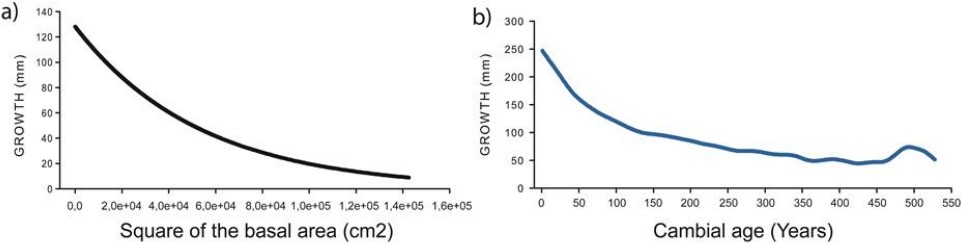

Figure 4. a) Represents the model of the BasPois method, b) represents the regional curve of
the RCS method.

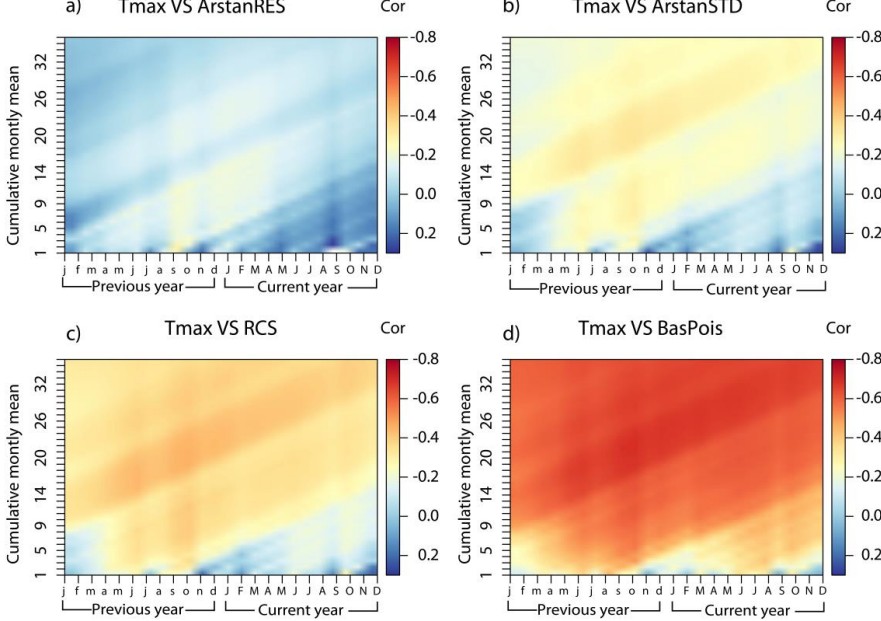

Figure 5. Correlation between the maximum temperature (from January of the previous year
to December of the current year with a cumulative monthly mean from 1 to 36 months) and
the residual Arstan chronology (a), the standard Arstan chronology (b), the RCS standard
chronology (c) and the Basal Area-Poisson standard chronology (d).



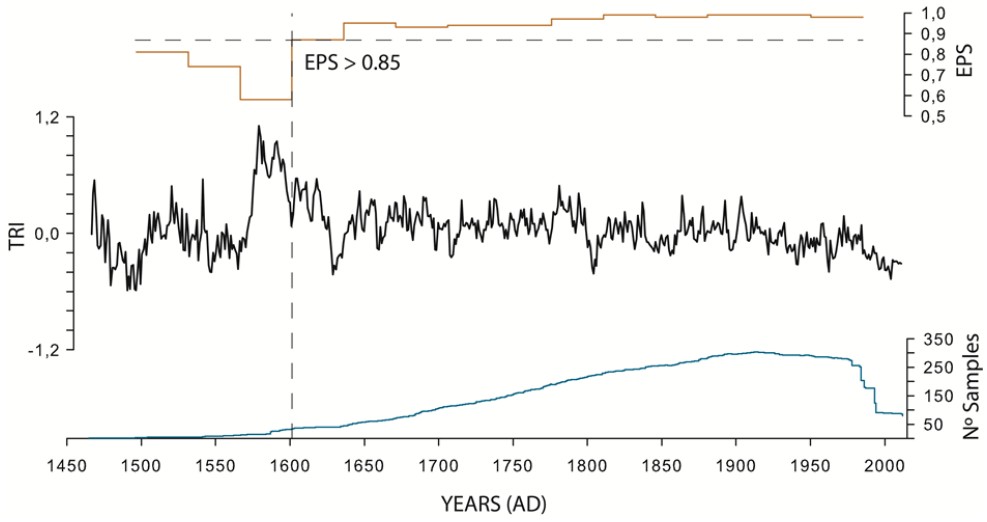

Figure 6. BasPois chronology (in black), number of samples (blue) and EPS statistic
(computed over 30-y window lagged by 15 years) back to 1465. Vertical dashed line
highlights the EPS=0.85 threshold in 1602.

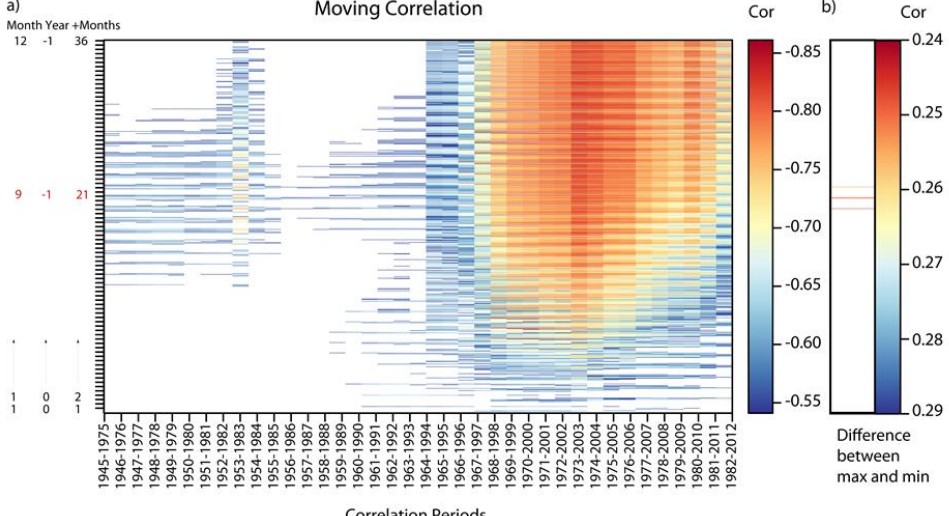

Figure 7.a) 30-year moving correlation from 1945 to 2012 between the maximum
temperature, from January of the current year (1,0,1) to December of the previous year (12, -
1, 36) with a cumulative monthly mean from 1 to 36 months and the BasPois chronology. Red
numbers indicates the chosen climatological parameter; 9, September, -1, previous year, 21,



1   months used for the cumulative monthly mean. b) The four best parameters are represented.

2   Reddish line indicates the least difference between the maximum and minimum correlation in

3   the correlation periods.

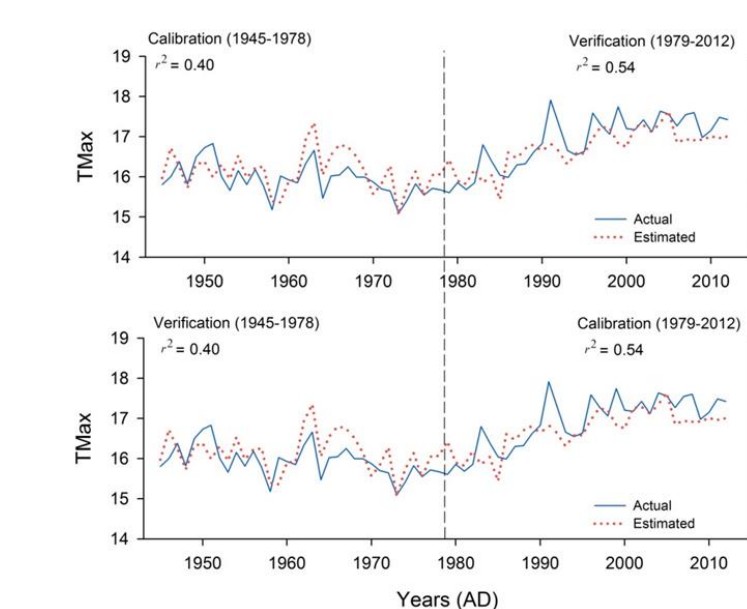

20   Figure 8. Calibration and verification results of the CRU data based $Tmax_{Sep-1}$ reconstruction




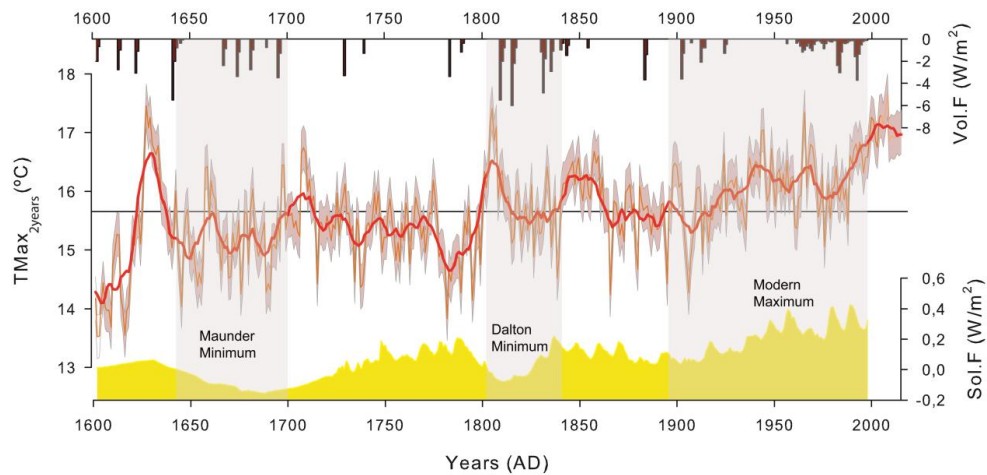

Figure 9. IR2T$_{max}$ reconstruction since AD 1602 for the Iberian Range. Bold red curve is a 11-year running mean, purple shading indicates the mean square error based on the calibration period correlation. Yellow shading at the bottom show solar forcing and bars on top indicate volcanic forcings (Crowley 2000).

|  | Calibration 1945-1978 | Verification 1978-2012 | Calibration 1979-2012 | Verification 1945-1978 | Period 1945-2012 |
|---|---|---|---|---|---|
| Years | 34 | 34 | 34 | 34 | 68 |
| Correlation | -0.64 | 0.73 | -0.74 | 0.64 | -0.78 |
| $R^2_{adj}$ | 0.40 | 0.54 | 0.54 | 0.40 | 0.61 |
| MSE | 0.09 | 0.66 | 0.18 | 0.29 | 0.37 |
| Reduction of error | 0.99 | 0.99 | 0.99 | 0.99 | 0.99 |
| Sing test | 28+/6- | 24+/10- | 28+/6- | 24+/10- | 52+/16- |

Table 2. Calibration/verification statistics of the Tmax$_{Sep-1}$ reconstruction





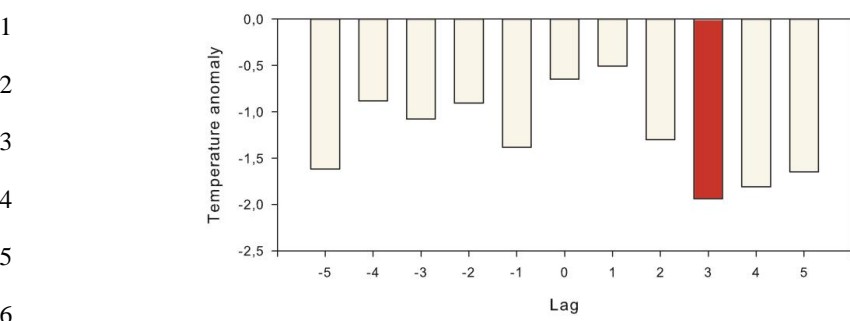

Figure 10. Superposed epoch analysis with a back and forward lag of 5 years. Significance ($p<0.05$) at 3 years after the extreme volcanic event.

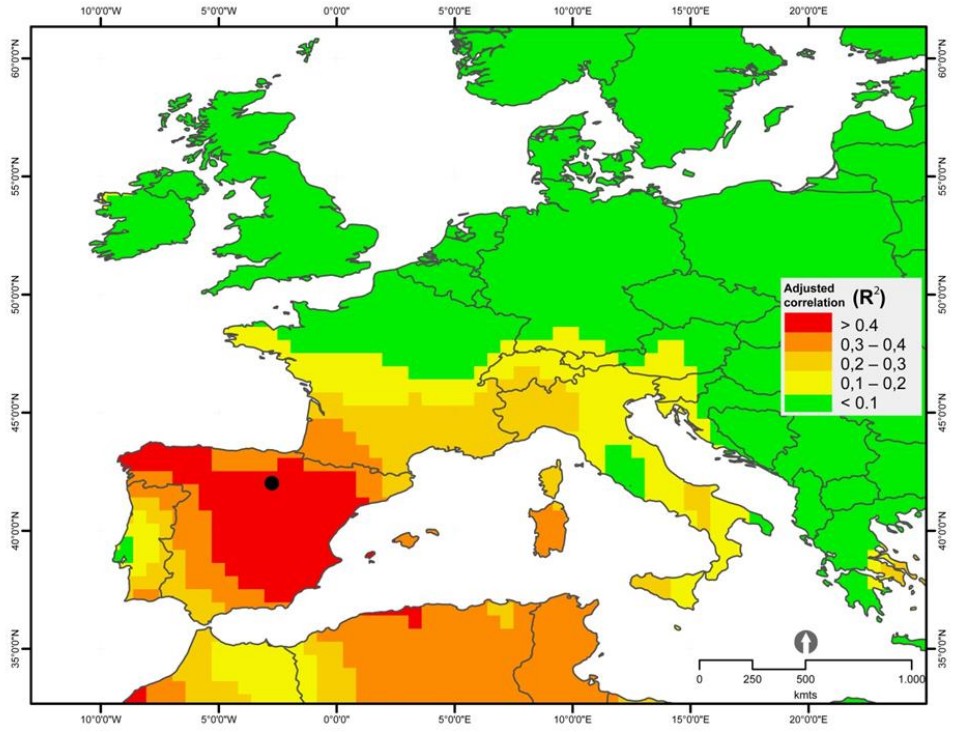

Figure 11. Map showing the spatial correlation patterns of the BasPois chronology with the gridded September of the previous year with a cumulative monthly mean of 21months data. Correlation values are significant at $p<0.0001$.