# Peer review of "Temperature variability of the Iberian Range since 1602 inferred from tree-ring records"

_Climate of the Past, 2016_

## Referee Comment (RC1) · Anonymous Referee #1 · 12 Feb 2016

In this study, a set of tree-ring chronologies from the Iberian Range is used to develop a maximum temperature reconstruction spanning the period 1602-2012. This topic is potentially very interesting since the temperature reconstructions in this region are rare. However, I see relevant issues that raise a number of (serious) concerns related to the composite chronology used for the reconstruction, the climate variable reconstructed and, particularly to the statistics of the calibration-verification. Considering these concerns, I unfortunately cannot recommend this manuscript for publication, and I think that addressing these concerns would entail the preparation of a whole new manuscript.

I will just focus on those main issues starting from the statistics of the calibration-verification. In addition, the manuscript would also require a careful editing since there are spelling errors, repetitions and inaccuracies, particularly related to the definition of

correlations (r) and coefficient of determination (r2).

I loathe to be so critical but Figure 8 and Table 2 give the impression that the numbers provided do not match with the series shown in the Figure and likely something went wrong when calculating and interpreting some statistics.

1. As an example, Figure 8 shows that the r2 of the later period is 0.54 (or a correlation of 0.73). This value does not seem to match the poor interannual synchrony between the series that can visually be seen in the figure. It seems to me that either the correlation is spurious and largely inflated by the similar trend; or a correlation of 0.54 was mistakenly labelled as r2. Please note that correlation (r) and coefficient of determination (r2) are used in the manuscript and figures in both upper and low case letters and sometimes mixed (i.e., in Figure 11. R2 is defined as adjusted correlation; and text between lines 27-31 in page 8 mention correlations but show values labelled as r2) and I wonder if this could has been a potential source of confusion.

2. The validation statistics seem also too high. The reduction of error (RE) value of 0.99 is just hard to believe. A RE value of 0.99 (considering that the theoretical maximum value is 1) would basically mean that trees are recording temperature with the precision of a thermometer and, unfortunately, this is not realistic. It is very likely that something went wrong in the calculation, and this would need to be re-checked and re-interpreted.

3. The reconstructed climate variable is the mean temperature over 21 months. This variable will presumably have a strong autocorrelation. It is not clear to me whether and how the authors statistically addressed the calculation of the significant levels considering the reduction in the degrees of freedom associated to a high autocorrelation. On the other hand, the authors stated on the manuscript that the chronology used for the reconstruction (BasPois) displays a first-order autocorrelation of 0 which implies that the proxy record does not mimic the autocorrelation of the temperature series used for calibration. Hence, there is a clear mismatch in the statistical properties of the predictor and the predictand. At this point, I am missing an analysis of the residuals from the

regression modelling (trend, autocorrelation, etc) that will provide critical information on the adequacy of the predictand included in the model. I would expect that the residuals derived from such a regression will show a strong autocorrelation which will question the estimations of the uncertainties and statistical significance.

4. In view of the clear visual mismatch in high frequency between the tree-ring chronology and the instrumental temperature record, I would recommend to do a comparison of both series at different time scales to make sure that the correlation observed in the calibrations are due to synchrony in both, low and high-frequency domains, and it is not a spurious correlation due to similar long-term trends. This would definitely help to know if the climate variable chosen for the calibration is the correct one.

In the background there are a couple of other issues that are not as relevant as the one with the statistics of the calibration-verification, but also critical in the general context of the paper.

5. The authors combined data from different chronologies into a single sort of regional chronology using different methods, which is always an interesting exercise. However, having a look to Figure 3, I wonder why all chronologies have been included into the final regional composite instead of discarding the chronologies that clearly showed poor correlations (i.e., s047). According to the information currently available on the paper, a reader cannot be sure whether chronologies encoding different climate signals have been merged into a final composite. To answer this question and reinforce the methodological decision adopted, I would suggest to check whether all chronologies encode the same climate signal before building regional chronologies, particularly if some chronologies clearly show a limited agreement with the rest. In this way, potential doubts on the quality and regional representativeness of the composite regional chronology will be minimized.

6. The link between the climate variable reconstructed in the paper and the proxy record lacks a consistent physiological explanation. The explanation given between

lines 13-16 of page 11, though correct in general terms, is too general and seems insufficient to explain the selection of a climate season that is quite unusual in the context of tree-ring based climate reconstructions. In fact, the explanation given could be applied to any lagged climate season. However, and independently of the physiological explanation, calibrating with a 21 cumulative monthly mean of temperature when the chronology shows a first order autocorrelation of 0, seems totally contradictory to me. I do not doubt that the authors have a consistent reason for all the decisions adopted in the paper. However, the present version of the manuscript gives the impression that the selection of the composite chronology and the climate season used for the reconstruction were purely based on the highest correlation obtained, and all other considerations and potential implications were somehow overlooked.

---

## Referee Comment (RC2) · Anonymous Referee #2 · 25 Feb 2016

The manuscript presents climate reconstruction for the 1602-2012 period for previously less explored part of the Iberian Peninsula. The authors used a new standardization method based on the basal area. The applied detrending improved the quality of the reconstruction since the size-based standardization maximizes the common signal. The presented reconstruction fills the gap in climatic reconstructions in the area vastly affected by climate change where long term climatic records (and old trees) are very rare.

The first version of the ms contained numerous small errors. Some of them were already commented by the referee #1. The authors have responded to them. Their response is to my opinion adequate. The authors presented their response and submitted an improved version of the manuscript. The errors are now corrected, although they generally did not affect the message (result and conclusions) of the manuscript.

[Figure]

The chronology is to my opinion well constructed and I agree with the response of the authors. The fact that they used different species has been adequately justified already in the first version of the ms. This is a common praxis if the chronologies match well - this is the case in this ms.

Both versions of the ms contain small errors (mainly language and style) which should be corrected at the end of the interactive review process.

———————————————————

---

## Author Comment (AC1) · 25 Feb 2016

Thank you for the comments. About the calibration-verification statistics, you are right, some of the values included in the submitted version of the manuscript were wrong (sorry about that). We perhaps should have delved deeper into the development of the chronology and the climate variable reconstructed. Through this comment we aim to answer all the questions that the manuscript has generated.

1. Regarding Figure 8, the correlation was not mistakenly labelled as r2. However, as suggested by the referee, we have correctly labelled the Pearson correlation (r) and the coefficient of determination (r2). The inter-annual synchrony that can be seen between the series denotes that the reconstruction is better at mid to low frequencies than at high frequency.

[Figure]

2. Regarding the calibration/verification statistics, we apologize for the error, the included in the ms were calculated using unstandardized series. The revised are now shown in Table2: RE for the period 1945-2012 is 0.56, so substantially lower than reported, but still indicating reconstruction skill.

3. Regarding autocorrelation, the correct value is 0.83, which is crucial for the development of a reconstruction retaining information of the past 21 months. To further assess the accuracy of the model we included a new figure (Model_Residuals) detailing the transfer model and regression residuals.

4. Regarding visual mismatch in the high frequency domain. Similar long-term trends between temperature and tree-rings are not necessarily indicative of a spurious relation, but might simply suggest that trees are, responding to long-term temperature trends. Sure this is difficult to assess due to limited degrees of freedom. However, preserving such trends remains a key challenge in tree-ring based climate reconstructions (Briffa et al. 1992, Esper et al. 2003b). We employed a running correlation analysis (Fig. 7) not only to test temporal stability, but also to support the selection of climate variable.

5. Regarding the regional chronology, we develop this timeseries by combining 11 sites including two pine species within an area of 90 square kilometers ranging from 1,500 to 1,900 masl. A new column in figure 3 showing the correlation between the single sites and the regional chronology provides perhaps useful information. Despite local differences among the sites, the group of chronologies shares common variance, and the mean chronology contains a clear climate signal. Data integration from this tree-ring network enabled the development of a regional rather than local reconstruction. Figure 11 shows the spatial extent of the reconstruction indicating r2 > 0.4 for the central and Mediterranean regions of the Iberian Peninsula.

6. Regarding physiological explanation. Extended between lines 15-31 of page 11 in the new version of the ms.

Please also note the supplement to this comment:
http://www.clim-past-discuss.net/cp-2016-9/cp-2016-9-AC1-supplement.pdf

———————————————————

|  | REGIONAL CHRONOLOGY | VIN(1900) | CAV(1900) | NEI(1850) | s048(1840) | s049(1840) | s047(1750) | s050(1750) | URB(1750) | s006(1630) | COV(1500) | HER(1500) |
|---|---|---|---|---|---|---|---|---|---|---|---|---|
| REGIONAL CHRONOLOGY | 1 | 0.47 | 0.65 | 0.69 | 0.69 | 0.74 | 0.69 | 0.63 | 0.83 | 0.7 | 0.83 | 0.71 |
| VIN(1900) | 0.6 | 1 | 0.83 | 0.33 | 0.18 | 0.17 | 0.26 | 0.33 | 0.42 | 0.04 | 0.27 | 0.34 |
| CAV(1900) | 0.73 | 0.83 | 1 | 0.37 | 0.29 | 0.28 | 0.42 | 0.41 | 0.55 | 0.3 | 0.46 | 0.4 |
| NEI(1850) | 0.74 | 0.51 | 0.56 | 1 | 0.59 | 0.7 | 0.39 | 0.42 | 0.53 | 0.24 | 0.61 | 0.57 |
| s048(1840) | 0.74 | 0.32 | 0.49 | 0.49 | 1 | 0.46 | 0.37 | 0.34 | 0.56 | 0.4 | 0.72 | 0.67 |
| s049(1840) | 0.57 | 0.13 | 0.23 | 0.58 | 0.39 | 1 | 0.48 | 0.54 | 0.7 | 0.4 | 0.53 | 0.57 |
| s047(1750) | 0.64 | 0.08 | 0.06 | 0.36 | 0.11 | 0.31 | 1 | 0.6 | 0.67 | 0.37 | 0.61 | 0.55 |
| s050(1750) | 0.65 | 0.4 | 0.41 | 0.45 | 0.44 | 0.3 | 0.39 | 1 | 0.55 | 0.23 | 0.47 | 0.44 |
| URB(1750) | 0.7 | 0.37 | 0.34 | 0.45 | 0.42 | 0.34 | 0.55 | 0.51 | 1 | 0.41 | 0.69 | 0.66 |
| s006(1630) | 0.7 | 0.04 | 0.3 | 0.24 | 0.4 | 0.4 | 0.37 | 0.23 | 0.41 | 1 | 0.54 | 0.25 |
| COV(1500) | 0.75 | 0.12 | 0.2 | 0.5 | 0.53 | 0.48 | 0.8 | 0.56 | 0.68 | 0.54 | 1 | 0.62 |
| HER(1500) | 0.69 | 0.41 | 0.39 | 0.47 | 0.54 | 0.45 | 0.39 | 0.53 | 0.6 | 0.25 | 0.56 | 1 |

**Fig. 1.**

[Figure]

**Fig. 2.**

[Figure]

**Supplement:**

[revised manuscript text omitted]

---

## Author Comment (AC4) · 25 Feb 2016

The comment was uploaded in the form of a supplement:
http://www.clim-past-discuss.net/cp-2016-9/cp-2016-9-AC4-supplement.pdf

————————————————

---

## Short Comment (SC1) · 1 Mar 2016

While the authors have addressed the remarks of Reviewer #1 in a reasonable and adequate way, I see some methodological problems, mainly with the RCS (baspois) application:

1: Did you use pith offset (or for your case of Basal-Area-RC distance to pith) estimates? I cannot find it in the text. If not, why? Omitting pith offset estimates will lower your RC and ultimately introduce a fake negative trend in the early years of your chronology. In your case of inversion a positive biased trend, which could be amplified when using Basal-Area. See Briffa & Melvin 2011 ∼"A closer look on RCS..." and Klesse & Frank 2013 (attached).

2. You include old ITRDB datasets from the 1980s. Do you have the samples or pith

offset estimates, or at least correct for the a- and b-sample difference of starting year? For example: a- and b-samples of the ITRDB series SPAI047 have quite large differences between their starting years (mean: 33 years). For their RCS curve that would mean, that those samples are overestimated on average already by 50mm (should be probably 0.5mm).

2b. The y axis in figure 4b is presumably off by a factor of 100 and should range from zero to 2.5mm instead of 250mm.

3. Do the trees have the same growth rates at all 11 sites? If not then a use of a single RC might introduce false trends, when sample and site replication changes. From originally 11 sites, 5 drop out in the 1980s, 4 in the 90s and you are left with only two sites. Do these sites have the same growth level as the ones that drop out before? See also Figure 6 in Klesse & Frank for an example of falsely introduced trends.

I have attached a figure showing this potential problem including the 5 Iberian Range (IR) ITRDB chronologies and 2 chronologies from Büntgen near Madrid. Although 250km to SW they grew at similar elevation and correlate with the mean chronology of the other 5 series quite good (r=0.52, 1701-1985, 30-year spline detrended). So, well in the range of your observed site to site correlations and only a little bit weaker than your weakest site to regional chronology corrlelation, but completely independent (one could actually argue to include them to increase the regional representation, but that's beside the point here). I applied a single RC and no pith offsets, split the IR and Büntgen series and averaged them separately with an arithmetic mean. It is obvious that the mean of Büntgen have permanently lower values over the IR series. So if the IR series drop out, the overall RCS chronology gets heavily drawn towards lower values, while the Büntgen series remain ±constant. This effect could probably also be enlarged using the size/basal area detrending.

Can you show that this does not cause a problem in your data?

4. Figure 5a) Why do you compare your residual AC-free chronology with raw temperature data? That does not make sense if you want to highlight common signal in the high-frequencies. The simplest method would have been to detrend both series with a flexible spline (e.g. 30 years). That actually comes back to Remark 4 from Reviewer #1. If the TRW signal is truly representing pSep21 temperature, than it still should at least have reasonable negative correlations on the high-frequencies. A running correlation with raw temperature and BasPois does not answer Remark 4 and still contains trend-in-signal and not necessarily causal effect.

I believe the authors might have kept things too easy during the RCS application, which might have lead to erroneous conclusions. I would be really happy if my concerns don't have an impact on the conclusions, but without showing that I remain cautious.

Please also note the supplement to this comment:
http://www.clim-past-discuss.net/cp-2016-9/cp-2016-9-SC1-supplement.pdf
* * *
[Figure]

[Figure]

**Fig. 1.**

**Supplement:**

**Testing the application of Regional Curve Standardization to living tree datasets**

**S. Klesse & D. Frank**

Swiss Federal Research Institute WSL, Birmensdorf, Switzerland Email: stefan.klesse@wsl.ch

**Introduction**

How to detrend tree-ring series is a crucial point in dendrochronology. There are many different approaches to remove age-related trends and unfortunately none of them can objectively answer the question, which is the best. It is the opinion of the individual researcher whether to choose a stochastic, a deterministic, or an empirical method, not to mention the plethora of options within each of these broad categories (Cook & Kariukstis 1990).

The classical deterministic detrending method fits a modified negative exponential curve or straight line to the tree-ring series. Stochastic methods use low-pass filter such as the cubic smoothing spline with a fixed or relative 50% frequency cut-off of a certain wavelength (Cook & Peters 1981). The so-called regional curve standardization (RCS) is a commonly applied empirical based upon the presumption that for a given species and site a common age related biological growth signal exists, independent of when the trees were growing.

With the task to reconstruct long-term climate variation in mind, Cook et al. (1995) note that "the maximum length of recoverable climate information is ordinarily related to the lengths of the individual tree-ring series" and usually ranges around a third of the mean segment length. To overcome this so-called "segment length curse" Briffa et al. (1992) re-popularized the empirical RCS method (Mitchell 1967). RCS allows for the possibility of systematic over or underestimation of the actual tree growth level during any particular time period due to changing environmental conditions and thus permits climatic information to be preserved on time-scales longer than the individual segment lengths (Cook et al. 1995). As tree-ring data are one of the most important data sources to place recent climate trends in a long-term context, researchers have heavily favored using RCS for climate reconstruction purposes during the past decade or so (Frank et al. 2010).

In RCS, all core segments are aligned by their cambial age and averaged to obtain the regional growth curve (RC). The RC is further smoothed, either by deterministic or stochastic means, to reduce high frequency variance and then applied to remove the age trend of each single series. After the regional curve derivation the detrended single series are reset to their calendar age. With this detrending method the century-scale fluctuations may be successfully preserved, however the correlation between individual samples decreases leading to a greater uncertainty than using "traditional" methods, thus a high sample replication is needed (Briffa et al. 1992).

Other more subtle challenges and limitations with RCS have only recently been recognized. With RCS, care has to be taken that one might even lose low-frequency when all samples span the full length of the chronology, because the overall climate signal is contained in the regional curve and therefore completely removed by standardization (Briffa & Melvin 2011). Thus ideally RCS is applied to composite datasets including trees with a uniform distribution of both germination and death (or sampling) dates. Yet, in practice such datasets are rare, and the desire to preserve long-term climate information from sites with only living trees is desired.

Briffa & Melvin (2011) recently described three major possible biases in applying RCS with living only data:

i) The "trend-in-signal" bias, which occurs when the growth-forcing signal has variance on timescales close to or greater than the length of the chronology. During detrending the average slope of each sample is removed and therefore distortions of the chronology, especially towards

the beginning and the end, may occur. In most cases this bias is of more minor concern since the existence of a trend overlying the chronology is unknown.

ii) The "different-contemporaneous-growth-rate" bias describes the problem if a single RC systematically differs from the age trend of fast or slow-growing trees, due to differences in nonclimatic factors such as exposition, soil quality, or competition. If in one period fast-growing trees outnumber the slow ones (or vice versa), false medium-frequency trends might become apparent (Briffa & Melvin 2011).

iii) The combination of variations in the longevity of trees, sampling practice and different growth rates of contemporaneous trees can lead to the "modern-sample" bias. Assuming that large trees have a higher risk of mortality trees are more likely to be killed by an extreme event when they are close to their maximum size. A slow growing tree takes longer to reach this size and therefore can become older (Black et al. 2008, Bigler & Veblen 2009). This bias becomes most obvious in chronologies with only living samples from some old, yet primarily young trees. The regional curve may overestimate the old and slow-growing trees at the beginning of the chronology and underestimate the overall chronology in the later years and hence lead to an increased rise of chronology indices, where the new young and faster growing trees contribute to the RC (Briffa & Melvin 2011).

Other challenges related to RCS application include the use of a single RCS curve to detrend data from clearly differing sites (e.g. Esper et al. 2002). Measurements at a specific site may be systematically under- or overestimated, due to the different growth levels at these particular sites with the bias depending upon the differences in chronology length and age structure among the sites. It is therefore required to normalize those single site chronologies before averaging them together to mitigate trends evolving due to changing site replication. A final recognized systematic bias in RCS curves may arise from not having information (i.e. pith offset (PO) estimates) on the quantity of rings missing between the innermost sampling ring and the stem pith. This absence will "reduce the expected ring width maximum in early years of tree growth and consequently lower the expected trend of declining growth with increasing age" (Briffa & Melvin 2011). The use of pith-offset estimates (POE) is generally recommended, as it will increase the accuracy of the growth curve.

In this study we focus on applying regional curve standardization (RCS) to samples from only living *Pinus nigra* from Mt. Olympus, Greece. The dataset consists of seven potentially drought sensitive sites and 556 samples, collected with the aim to reconstruct past summer moisture variability. The sites span the lower to upper elevation range of black pine forest in this area (850m – 1700m) and are distributed around the Mt. Olympus massive to capture different slope exposures and luff-lee conditions. We only use latewood width (LWW) measurements, as a previous investigation (Klesse 2012) revealed that this parameter contains the strongest response to summer drought conditions (May-July). We aim to assess and ideally overcome challenges related to employing RCS on this dataset with the ultimate aim to reconstruct inter-annual to multi-centennial climate variability. Accordingly, all site chronologies are screened for the above mentioned possible biases, the steps to construct the best possible master chronology are explained, and a preliminary May-July Standardized Precipitation Index (SPI) reconstruction for the last 400 years is shown.

**Testing for biases in the Mt. Olympus dataset**

Fig. 1a shows the seven TRW chronologies, detrended with an individual RC (smoothed with a 50year spline) fitted for each site. Great similarities in the lower-frequency behaviour of the three old sites (PPP, PIGA and XEP) including a decreasing trend from the second half of the 17th century until the end of the 19th century and a strong upward shift (1895-1912) at the turn of the century. During the period of overlap the younger LIA and VPA sites shares this common low-frequency variability, where as the two other sites (REB and CHR) agree less well in the low-frequency domain.

Figure 1: a) Seven individually detrended  $RCS_{TRW}$  chronologies from Mt. Olympus, Greece and b) showing pith offset estimates of 556 individual cores from all seven sites as a function of the calendar year of the innermost measured ring.

The PO distribution is generally homogeneous over time, with the exception that high PO values above 100 years accumulate in the 18th and 19th century (Fig. 1b). These high POs primarily arose while truncating some samples that could not be confidently crossdated due to their extremely low growth and many missing rings. The latewood RC from all sites combined without PO is shifted approximately 15 years towards the juvenile phase (Fig. 2a), but has the same slope. While the RCs are quite similar, in practice the innermost rings of a sample with a 150 year PO would be divided by ~0.2, whilst if the PO data were not considered the detrending starts with values over 0.4 - a huge difference. In this study only about 20 of 556 samples have large PO and their consideration tends to have a rather small influence. Differences are most extreme at the PPP site for TRW, where the slope without POE is much shallower than with POEs (Fig. 2b). Most of the segments start within the first 200 years of the chronology (Fig. 2c). If the POs are disregarded growth of those samples with a small PO (

Figure 2: Comparisons of regional curves with (black) and without pith offset estimates (POE) (grey) with a) showing the differences using all 556 LWW samples and **b**) the extreme case of TRW at the PPP site. **c**) temporal distribution of the pith offset estimates for the PPP site and **d**) the resulting RCS chronologies with and without the use of POEs (black and grey, respectively) for PPP.

---

## Referee Comment (RC3) · Anonymous Referee #1 · 8 Mar 2016

The effort of the authors commenting and answering my questions and assuming mistakes in the calculation of some statistics is laudable. However, relevant issues described in detail in my initial review have not been answered and the reliability of the calibration is still questionable. The key component of this manuscript is the development of a climate reconstruction based on tree rings. If the calibration raises serious doubts, then the whole manuscript is dubious. I am aware of the effort that is needed to develop a proxy-based climate reconstructions, however, I still believe that the reconstruction presented in this manuscript is not fully reliable mainly due to three main issues:

1. How good or bad is the agreement between the tree-ring record and the climate record on the high-frequency domain remains unanswered. A poor visual agreement on the high frequency is obvious having a look to Figure 8, and the author's reply

("the reconstruction is better at mid and low frequencies than at high frequencies") is insufficient and does not provide any additional or new information to clarify this point. The way to solve the doubt is, as suggested in point 4 of my initial review and also suggested by S.Klesse, to do a comparison (correlation) of both series at different time scales and make sure that the correlation observed in the calibration are due also to synchrony in the high frequency and not only to a similar long-term trend. If the series correlate on the low frequency but do not show a reasonable agreement on the high frequency, then the correlation shown for the calibration period would be spurious and the reconstruction simply incorrect. The running correlation analysis will only answer this question if the series would have been high-pass filtered, which is not the case. In addition, the residual analysis now included on the paper would also require to include some test on the trend and autocorrelation of the residuals (i.e., Durbin-Watson test). If the calibration fulfils all above (agreement on the high-frequency domain and test of residuals), then we could talk about a statistical reliable calibration.

2. Whether chronologies encoding different climate signals have been merged into a final composite remains also unanswered. The new column added to Figure 3 containing the values of the correlations between the single and regional chronologies does not answer my initial question. Checking whether all chronologies encode the same climate signal means to correlate each individual chronology with climate. This is the way to know if tree growth is limited by the same climatic factor at all sites or different climate signals are being mixed in the regional chronology. Considering that the chronologies used are from different tree species, derived from different elevations and some chronologies do show poor correlation with the others, testing potential different climate signals is advisable, particularly because solving such a question is extremely easy.

3. The physiological explanation is still too general (in fact, has not substantially changed) and not very convincing. It is hard for me to picture how tree growth can be negatively influenced by the cumulative mean of temperature from the current and

previous year of growth: how trees manage to grow then? How did they survive for centuries and did not die by carbon starvation if cumulative temperature of the previous 21 months have no positive effect on growth? Physiologically seems quite unlikely to me but still, I was hoping for a good explanation or answer that could challenge my thoughts on this regard.

---

## Author Comment (AC5) · 28 Mar 2016

Dear Stefan:

1. Regarding pith offset. On the one hand, we use a set of sites (VIN, CAV, NEI, URB, COV, HER) of which the samples are available, and therefore pith offsets can be estimated. On the other hand, we use a set of sites from the ITRDB (s047, s048, s049, s050, s006) from which the samples cannot be accessed. Here, for each tree, we have assumed PO = 1 in the oldest sample, and adjusted the shorter series accordingly. Sure this procedure introduces uncertainties, but this is true for all studies using data from the ITRDB. We believe, however, that these uncertainties do not generate a systematic bias, but are minimized using the new BasPois detrended method based on basal area instead of age.

[Figure]

2. Perhaps already addressed in the previous paragraph. Since we do not have the ITRDB samples, pith offsets need to be estimated. Assuming an age of 1 in the oldest sample of every tree (of the ITRDB data) is indeed a compromise, and perhaps there are other possible solutions, however, we believe that for the development of a chronology representative of the regional climate of the western Iberian Range, all available samples should be included.

2b. We apologize for the mistake. The axis is now correctly labelled (Fig.4).

3. About growth rates, the variability among sites is lower than the variability within a site. Besides, we are not joining chronologies, but develop the regional chronology from all 316 individual TRW series. It is generally unavoidable to add some noise when integrating TRW series from different locations and species in a regional chronology. However, the high correlation between each site and the regional chronology suggests a general climatic signal. Similar approaches have been detailed in Briffa et al. 1998 and numerous climatic reconstructions have been developed using networks of different sites and species i.e. Wilson et al. 2003, Battipaglia et al. 2010, Büntgen et al. 2011 or, Esper et al. 2012. However, to prove that the variable end dates are without effect in the trend we develop a regional chronology with the sites ending in 1993 (using the BasPois detrended method) (Fig_1).

4. As suggested, we have detrended the climate data using a flexible spline (30 years) and correlated with ArstanRES and ArstanSTD chronologies emphasizing high frequency variance. The results show an increase in correlation with pSep21 temperature; for ArstanRES the correlation is $r=-0.39$, while for ArstanSTD the correlation increases to $r=-0.56$. These correlations indicate that the reconstruction contains some skill in the high frequency domain. Nevertheless, in order to assess long-term climate variability, we prefer using the BasPois chronology, in which the climate signal is enhanced, and both high and low frequency pSep21 variance and forcing (volcanic and solar) is retained. The changes in methodology and results have been included in the manuscript as well as the new Figure 5 (here Fig2).

Please also note the supplement to this comment:
http://www.clim-past-discuss.net/cp-2016-9/cp-2016-9-AC5-supplement.pdf
* * *
[Figure]

[Figure]

**Fig. 1.**

a)  Tmax VS ArstanRES

b)  Tmax VS ArstanSTD

c)  Tmax VS RCS

d)  Tmax VS BasPois

Fig. 2.

**Supplement:**

[revised manuscript text omitted]

1842-1977. Bottom left shows the correlation over the full period of overlap between pairs of chronologies

[Figure]

Figure 4. a) Represents the model of the BasPois method, b) represents the regional curve of the RCS method.

[Figure]

Figure 5. Correlation between the maximum temperature (from January of the previous year to December of the current year with a cumulative monthly mean from 1 to 36 months) and the residual Arstan chronology (a), the standard Arstan chronology (b), the RCS standard chronology (c) and the Basal Area-Poisson standard chronology (d).

[Figure]

Figure 6. BasPois chronology (in black), number of samples (blue) and EPS statistic (computed over 30-y window lagged by 15 years) back to 1465. Vertical dashed line highlights the EPS=0.85 threshold in 1602.

[Figure]

Figure 7.a) 30-year moving correlation from 1945 to 2012 between the maximum temperature, from January of the current year (1,0,1) to December of the previous year (12, -

1, 36) with a cumulative monthly mean from 1 to 36 months and the BasPois chronology. Red numbers indicates the chosen climatological parameter; 9, September, -1, previous year, 21, months used for the cumulative monthly mean. b) The four best parameters are represented. Reddish line indicates the least difference between the maximum and minimum correlation in the correlation periods.

[Figure]

Figure 8. Calibration and verification results of the CRU data based $Tmax_{Sep-1}$ reconstruction

[Figure]

Figure 9. IR2T$_{max}$ reconstruction since AD 1602 for the Iberian Range. Bold red curve is a 11-
year running mean, purple shading indicates the mean square error based on the calibration
period correlation. Yellow shading at the bottom show solar forcing and bars on top indicate
volcanic forcings (Crowley 2000).

|  | Calibration 1945-1978 | Verification 1978-2012 | Calibration 1979-2012 | Verification 1945-1978 | Period 1945-2012 |
|---|---|---|---|---|---|
| Years | 34 | 34 | 34 | 34 | 68 |
| Correlation | -0.64 | 0.73 | -0.74 | 0.64 | -0.78 |
| R$^2$ | 0.41 | 0.55 | 0.55 | 0.41 | 0.61 |
| MSE | 0.43 | 0.42 | 0.42 | 0.43 | 0.86 |
| Reduction of error | 0.40 | 0.65 | 0.65 | 0.40 | 0.56 |
| Sing test | 28+/6- | 24+/10- | 28+/6- | 24+/10- | 52+/16- |
| Durbin-Watson | 1.31 $p<0.01$ | 1.53 $p<0.05$ | 1.53 $p<0.05$ | 1.31 $p<0.01$ | 1.45 $p<0.001$ |

Table 2. Calibration/verification statistics of the Tmax$_{Sep-1}$ reconstruction

[Figure]

Figure 10. Superposed epoch analysis with a back and forward lag of 5 years. Significance
($p<0.05$) at 3 years after the extreme volcanic event.

[Figure]

1 Figure 11. Map showing the spatial correlation patterns of the BasPois chronology with the

2 gridded September of the previous year with a cumulative monthly mean of 21months data.

3 Correlation values are significant at p<0.0001.

---

## Author Comment (AC6) · 28 Mar 2016

Referee #1: Thank you for your interest and comments, which we aim to answer here.

1. In order to test the agreement between the tree-ring chronology and climate record in the high-frequency domain, and in line with Stefan Klesse's suggestions, we correlated the ArstanRES and ArstanSTD timeseries with the detrended (30-year spline) climatic data. The results show an increase in correlation with pSep21 temperature. For ArstanRES the correlation is r= -0.39, whereas for ArstanSTD the correlation increases to r=-0.56. These results validate correlation in the high-frequency domain and indicate that the reconstruction signal is not spurious. However, we intend to reconstruct both high and low frequency climate variations and prefer using the BasPois chronology as it enhances the climatic signal (r=-0.78) and reproduces the full variance spectrum of the pSep21 variable very well. We performed a Durbin-Watson test, as suggested, to assess residual autocorrelation. The results were added to Table2. The Durbin Watson value for the period 1945-2012 is 1.45 (p<0.001) indicating no substantial autocorrelation in the residuals.

We believe that both the correlations after removal of low frequency variance as well as the insignificant autocorrelation in the residuals support the pSep21 reconstruction.

2. About chronology development, we did not merge site chronologies, but applied the standardization methods to all 316 individual TRW series to produce a regional chronology. Nonetheless, we added climate calibrations for each site to validate that the climate signal is regionally consistent. We developed a chronology for each of the 11 sites (detrended with the BasPois method) and correlated with the climatic variables. Highest correlations in the 11 sites appear for pSep20, pSep21 and pOct21. Since we chose to reconstruct pSep21 we also performed running correlations using a 30-year window to assess correlation stability within the calibration period. Results are shown in the Fig.1 and chronologies are sorted by elevation, VIN and CAV are Pinus uncinata, while the rest are Pinus sylvestris. The correlation never drops below r= -0.2. There are also periods surpassing r=-0.80. However, we would like to reemphasize that the aim of this study it is not to develop a local climate reconstruction, but to reconstruct the regional climate of the western Iberian Range.

3. We would like to remark that tree-ring growth it is not negatively influenced by temperature. It is, however, negatively correlated with temperature of the previous year using a cumulative monthly mean of 21 months. That would mean that within the environment in which trees are growing and with respect to the mean, they will grow more in cold years than in hot years. The negative temperature correlation is already shown for the previous September (r=-0.56) without any cumulative monthly mean. This negative temperature correlation has been reported in numerous dendroclimatic studies (i.e. Büntgen et al. 2006 or van der Werf et al. 2007) including the most recently developed climatic reconstruction for the Iberian Peninsula by Dorado-Liñán et al. 2014 showing a negative correlation with previous summer temperatures. One of the strengths of this paper is precisely adding the cumulative monthly mean to the climate variables which maximizes the correlation to r=-0.78. The ecophysiological explanation of previous year's influence on current's year tree-ring growth was already related with the storage of starch and sugar in parenchyma ray tissue and the remobilization of carbohydrates from root structures. Memory effects on TRW data have also been studied regarding the delayed response in TRW to post volcanic eruptions (1âĹij5 years) associated with a decrease in current's year temperature (D'Arrigo et al., 2013, Esper et al., 2014).

We agree on the need to conduct further studies to better understand the full range of ecophysiological processes of pine and other species. To this extend, we are aware of an experiment conducted by a colleague (Dr. Eustaquio Gil Pelegrin; ttps://www.researchgate.net/profile/Eustaquio_Pelegrin) in which they try to demonstrate that the generation of pinecones and needles in pine trees is very slow and it generally takes two years.

Please also note the supplement to this comment:
http://www.clim-past-discuss.net/cp-2016-9/cp-2016-9-AC6-supplement.pdf

[Figure]

Moving Correlation between sites and
September of the previous year with 21 cumulative monthly months

Fig. 1.

**Supplement:**

[revised manuscript text omitted]

1842-1977. Bottom left shows the correlation over the full period of overlap between pairs of chronologies

[Figure]

Figure 4. a) Represents the model of the BasPois method, b) represents the regional curve of the RCS method.

[Figure]

Figure 5. Correlation between the maximum temperature (from January of the previous year to December of the current year with a cumulative monthly mean from 1 to 36 months) and the residual Arstan chronology (a), the standard Arstan chronology (b), the RCS standard chronology (c) and the Basal Area-Poisson standard chronology (d).

[Figure]

Figure 6. BasPois chronology (in black), number of samples (blue) and EPS statistic (computed over 30-y window lagged by 15 years) back to 1465. Vertical dashed line highlights the EPS=0.85 threshold in 1602.

[Figure]

Figure 7.a) 30-year moving correlation from 1945 to 2012 between the maximum temperature, from January of the current year (1,0,1) to December of the previous year (12, -

1, 36) with a cumulative monthly mean from 1 to 36 months and the BasPois chronology. Red numbers indicates the chosen climatological parameter; 9, September, -1, previous year, 21, months used for the cumulative monthly mean. b) The four best parameters are represented. Reddish line indicates the least difference between the maximum and minimum correlation in the correlation periods.

[Figure]

Figure 8. Calibration and verification results of the CRU data based $Tmax_{Sep-1}$ reconstruction

[Figure]

Figure 9. IR2T$_{max}$ reconstruction since AD 1602 for the Iberian Range. Bold red curve is a 11-
year running mean, purple shading indicates the mean square error based on the calibration
period correlation. Yellow shading at the bottom show solar forcing and bars on top indicate
volcanic forcings (Crowley 2000).

|  | Calibration 1945-1978 | Verification 1978-2012 | Calibration 1979-2012 | Verification 1945-1978 | Period 1945-2012 |
|---|---|---|---|---|---|
| Years | 34 | 34 | 34 | 34 | 68 |
| Correlation | -0.64 | 0.73 | -0.74 | 0.64 | -0.78 |
| R$^2$ | 0.41 | 0.55 | 0.55 | 0.41 | 0.61 |
| MSE | 0.43 | 0.42 | 0.42 | 0.43 | 0.86 |
| Reduction of error | 0.40 | 0.65 | 0.65 | 0.40 | 0.56 |
| Sing test | 28+/6- | 24+/10- | 28+/6- | 24+/10- | 52+/16- |
| Durbin-Watson | 1.31 $p<0.01$ | 1.53 $p<0.05$ | 1.53 $p<0.05$ | 1.31 $p<0.01$ | 1.45 $p<0.001$ |

Table 2. Calibration/verification statistics of the Tmax$_{Sep-1}$ reconstruction

[Figure]

Figure 10. Superposed epoch analysis with a back and forward lag of 5 years. Significance
($p<0.05$) at 3 years after the extreme volcanic event.

[Figure]

1 Figure 11. Map showing the spatial correlation patterns of the BasPois chronology with the

2 gridded September of the previous year with a cumulative monthly mean of 21months data.

3 Correlation values are significant at p<0.0001.

---

## Short Comment (SC2) · 9 May 2016

After reading this discussion paper, I was left with the doubt on whether the authors have reconstructed temperature or precipitation. Considering that all or most of these sites are probably sensitive to variations in soil moisture given their location in Mediterranean mountains, at least a mixed precipitation-temperature signal could be expected and should be analyzed and discussed.

One must be extremely careful when analyzing negative effects of temperature on tree growth, particularly at sites >1500 m asl where temperatures are most likely not warm enough to cause direct damage to plant cells. What would be the biological mechanism of a 21-month long cumulative negative effect of temperature, if it were not for an indirect effect through hydric stress of the trees? It makes sense that these

relationships are driven by temperature increasing the evaporative demand or vapor pressure deficit. Thus, precipitation or a drought index should be considered in the analysis. I don't think this issue has been addressed in the original file or in the author's response to referee #1.

Results for SEA for volcanic eruptions would show lower temperature 3 years decrease after eruptions. That would mean wider tree rings. But those could also be caused by increased precipitation as shown for other parts of the Mediterranean Basin (Köse, N. et al., 2013. An improved reconstruction of May-June precipitation using tree-ring data from western Turkey and its links to volcanic eruptions. International Journal of Biometeorology, 57(5): 691-701.)

I would suggest the use of partial correlations for temperature (secondary variable), controlling for the effect of precipitation (primary variable). Using something like the seascorr function [Meko, D.M., Touchan, R. and Anchukaitis, K.J., 2011. Seascorr: A MATLAB program for identifying the seasonal climate signal in an annual tree-ring time series. Computers & Geosciences, 37(9): 1234-1241] should be straightforward. I would recommend the same time periods and lags be analyzed for precipitation or a drought index (similar to figure 5), before performing a temperature reconstruction from negative correlations with tree rings.

It may be that the correlations with temperature are ok, but I think this deserves better explanations and justifications.

---

## Referee Comment (RC4) · Anonymous Referee #3 · 6 Jul 2016

I find this paper intends to show and interesting matter, it is well written, and with a good quality of figures. Only some typos need to be corrected. However, there are some aspects that need clarification before the paper can be ready for publication, and some of them make me doubt about the validity of the results.

First of all, I wonder what the real objective of the paper is. On the one hand, different standardization methods are tested, and on the other hand the authors perform a reconstruction of a climatic variable. Though I can understand that this is a necessary step to provide a reliable reconstruction, I do not see that has this been brought in detail into the discussion, especially as regards the first two methods. But my main concern regards the variable selected for reconstruction. Though statistics seem to me optimal, I am not able to figure out what the causes for the existent relationship could be (21 month temperature). Explanations in the Discussion are too weak to be convincing,

and I think this aspect needs to be much better clarified or hypothesized.

---

## Author Comment (AC8) · 12 Jul 2016

Referee #3: Thank you for your interest and comments. The issues raised in your comments have been deeply treated throughout the open discussion process. We believe that in the latest version of the manuscript questions related with the standardization method and the 21 months climate variable have been clarified. In any case, we will be glad to address any particular aspect if there is a further suggestion. Attached is the latest version of the manuscript.

Please also note the supplement to this comment:
http://www.clim-past-discuss.net/cp-2016-9/cp-2016-9-AC8-supplement.pdf
* * *
[Figure]

**Supplement:**

[revised manuscript text omitted]

1842-1977. Bottom left shows the correlation over the full period of overlap between pairs of chronologies

[Figure]

Figure 4. a) Represents the model of the BasPois method, b) represents the regional curve of the RCS method.

[Figure]

Figure 5. Correlation between the maximum temperature (from January of the previous year to December of the current year with a cumulative monthly mean from 1 to 36 months) and the residual Arstan chronology (a), the standard Arstan chronology (b), the RCS standard chronology (c) and the Basal Area-Poisson standard chronology (d).

[Figure]

Figure 6. BasPois chronology (in black), number of samples (blue) and EPS statistic
(computed over 30-y window lagged by 15 years) back to 1465. Vertical dashed line
highlights the EPS=0.85 threshold in 1602.

[Figure]

Figure 7.a) 30-year moving correlation from 1945 to 2012 between the maximum
temperature, from January of the current year (1,0,1) to December of the previous year (12, -
1, 36) with a cumulative monthly mean from 1 to 36 months and the BasPois chronology. Red
numbers indicates the chosen climatological parameter; 9, September, -1, previous year, 21, months used for the cumulative monthly mean. b) The four best parameters are represented. Reddish line indicates the least difference between the maximum and minimum correlation in the correlation periods.

[Figure]

Figure 8. Calibration and verification results of the CRU data based Tmax$_{Sep-1}$ reconstruction

[Figure]

Figure 9. IR2T$_{max}$ reconstruction since AD 1602 for the Iberian Range. Bold red curve is a 11-
year running mean, purple shading indicates the mean square error based on the calibration
period correlation. Yellow shading at the bottom show solar forcing and bars on top indicate
volcanic forcings (Crowley 2000).

| | Calibration 1945-1978 | Verification 1978-2012 | Calibration 1979-2012 | Verification 1945-1978 | Period 1945-2012 |
|---|---|---|---|---|---|
| Years | 34 | 34 | 34 | 34 | 68 |
| Correlation | -0.64 | 0.73 | -0.74 | 0.64 | -0.78 |
| R$^2$ | 0.41 | 0.55 | 0.55 | 0.41 | 0.61 |
| MSE | 0.43 | 0.42 | 0.42 | 0.43 | 0.43 |
| Reduction of error | 0.40 | 0.65 | 0.65 | 0.40 | 0.56 |
| Sing test | 28+/6- | 24+/10- | 28+/6- | 24+/10- | 52+/16- |
| Durbin-Watson | 1.31 $p<0.01$ | 1.53 $p<0.05$ | 1.53 $p<0.05$ | 1.31 $p<0.01$ | 1.45 $p<0.001$ |

Table 2. Calibration/verification statistics of the Tmax$_{Sep-1}$ reconstruction

[Figure]

Figure 10. Superposed epoch analysis with a back and forward lag of 5 years. Significance (*p<0.05*) at 3 years after the extreme volcanic event.

[Figure]

Figure 11. Map showing the spatial correlation patterns of the BasPois chronology with the gridded September of the previous year with a cumulative monthly mean of 21months data.

Correlation values are significant at p<0.0001.

---

## Editor Comment (EC1) · V. Rath (Editor) · 8 Aug 2016

Dear Authors, the reviewers for this manuscript do not agree on the value of this manuscript. However I think you should submit a revised version to CP, where it will undergo further review. Best regards, Volker Rath
* * *

---

## Author Response (AR1)

**Interactive discussion on 'Temperature variability of the Iberian Range since 1602 inferred from tree-ring records'. Cp-2016-9**

**Anonymous Referee #1. 12 February, 2016.**

In this study, a set of tree-ring chronologies from the Iberian Range is used to develop a maximum temperature reconstruction spanning the period 1602-2012. This topic is potentially very interesting since the temperature reconstructions in this region are rare. However, I see relevant issues that raise a number of (serious) concerns related to the composite chronology used for the reconstruction, the climate variable reconstructed and, particularly to the statistics of the calibration-verification. Considering these concerns, I unfortunately cannot recommend this manuscript for publication, and I think that addressing these concerns would entail the preparation of a whole new manuscript. I will just focus on those main issues starting from the statistics of the calibration verification. In addition, the manuscript would also require a careful editing since there are spelling errors, repetitions and inaccuracies, particularly related to the definition of correlations (r) and coefficient of determination (r2). I loathe to be so critical but Figure 8 and Table 2 give the impression that the numbers provided do not match with the series shown in the Figure and likely something went wrong when calculating and interpreting some statistics.

**Thank you for the comments. About the calibration-verification statistics, you are right, some of the values included in the submitted version of the manuscript were wrong (sorry about that). We perhaps should have delved deeper into the development of the chronology and the climate variable reconstructed. Through this comment we aim to answer all the questions that the manuscript has generated.**

1. As an example, Figure 8 shows that the r2 of the later period is 0.54 (or a correlation of 0.73). This value does not seem to match the poor interannual synchrony between the series that can visually be seen in the figure. It seems to me that either the correlation is spurious and largely inflated by the similar trend; or a correlation of 0.54 was mistakenly labelled as r2. Please note that correlation (r) and coefficient of determination (r2) are used in the manuscript and figures in both upper and low case letters and sometimes mixed (i.e., in Figure 11. R2 is defined as adjusted correlation; and text between lines 27-31 in page 8 mention correlations but show values labelled as r2) and I wonder if this could has been a potential source of confusion.

**1. Regarding Figure 8, the correlation was not mistakenly labelled as r2. However, as suggested by the referee, we have correctly labelled the Pearson correlation (r) and the coefficient of determination (r2). The inter-annual synchrony that can be seen between the series denotes that the reconstruction is better at mid to low frequencies than at high frequency.**

2. The validation statistics seem also too high. The reduction of error (RE) value of 0.99 is just hard to believe. A RE value of 0.99 (considering that the theoretical maximum value is 1) would basically mean that trees are recording temperature with the precision of a thermometer and, unfortunately, this is not realistic. It is very likely that something went wrong in the calculation, and this would need to be re-checked and re-interpreted.

**2. Regarding the calibration/verification statistics, we apologize for the error; the included in the ms were calculated using unstandardized series. The revised are now shown in Table2: RE for the period 1945-2012 is 0.56, so substantially lower than reported, but still indicating reconstruction skill.**

3. The reconstructed climate variable is the mean temperature over 21 months. This variable will presumably have a strong autocorrelation. It is not clear to me whether and how the authors statistically addressed the calculation of the significant levels considering the reduction in the degrees of freedom associated to a high autocorrelation. On the other hand, the authors stated on the manuscript that the chronology used for the reconstruction (BasPois) displays a first-order autocorrelation of 0 which implies that the proxy record does not mimic the autocorrelation of the temperature series used for calibration. Hence, there is a clear mismatch in the statistical properties of the predictor and the predictand. At this point, I am missing an analysis of the residuals from the regression modelling (trend, autocorrelation, etc) that will provide critical information on the adequacy of the predictand included in the model. I would expect that the residuals derived from such a regression will show a strong autocorrelation which will question the estimations of the uncertainties and statistical significance.

**3. Regarding autocorrelation, the correct value is 0.83, which is crucial for the development of a reconstruction retaining information of the past 21 months. To further assess the accuracy of the model we included a new figure (Model_Residuals) detailing the transfer model and regression residuals.**

[Figure]

4. In view of the clear visual mismatch in high frequency between the tree-ring chronology and the instrumental temperature record, I would recommend to do a comparison of both series at different time scales to make sure that the correlation observed in the calibrations are due to synchrony in both, low and high-frequency domains, and it is not a spurious correlation due to similar long-term trends. This would definitely help to know if the climate variable chosen for the calibration is the correct one. In the background there are a couple of other issues that are not as relevant as the one with the statistics of the calibration-verification, but also critical in the general context of the paper.

**4. Regarding visual mismatch in the high frequency domain. Similar long-term trends between temperature and tree-rings are not necessarily indicative of a spurious relation, but might simply suggest that trees are, responding to long-term temperature trends. Sure this is difficult to assess due to limited degrees of freedom. However, preserving such trends remains a key challenge in tree-ring based climate reconstructions (Briffa et al. 1992, Esper et al. 2003b). We employed a running correlation analysis (Fig. 7) not only to test temporal stability, but also to support the selection of climate variable.**

5. The authors combined data from different chronologies into a single sort of regional chronology using different methods, which is always an interesting exercise. However, having a look to Figure 3, I wonder why all chronologies have been included into the final regional composite instead of discarding the chronologies that clearly showed poor correlations (i.e., s047). According to the information currently available on the paper, a reader cannot be sure whether chronologies encoding different climate signals
have been merged into a final composite. To answer this question and reinforce the methodological decision adopted, I would suggest to check whether all chronologies encode the same climate signal before building regional chronologies, particularly if some chronologies clearly show a limited agreement with the rest. In this way, potential doubts on the quality and regional representativeness of the composite regional chronology will be minimized.

**5. Regarding the regional chronology, we develop this timeseries by combining 11 sites including two pine species within an area of 90 square kilometers ranging from 1,500 to 1,900 masl. A new column in figure 3 showing the correlation between the single sites and the regional chronology provides perhaps useful information. Despite local differences among the sites, the group of chronologies shares common variance, and the mean chronology contains a clear climate signal. Data integration from this treering network enabled the development of a regional rather than local reconstruction. Figure 11 shows the spatial extent of the reconstruction indicating r2 > 0.4 for the central and Mediterranean regions of the Iberian Peninsula.**

[Figure]

6. The link between the climate variable reconstructed in the paper and the proxy record lacks a consistent physiological explanation. The explanation given between lines 13-16 of page 11, though correct in general terms, is too general and seems insufficient to explain the selection of a climate season that is quite unusual in the context of tree-ring based climate reconstructions. In fact, the explanation given could be applied to any lagged climate season. However, and independently of the physiological explanation, calibrating with a 21 cumulative monthly mean of temperature when the chronology shows a first order autocorrelation of 0, seems totally contradictory to me. I do not doubt that the authors have a consistent reason for all the decisions adopted in the paper. However, the present version of the manuscript gives the impression that the selection of the composite chronology and the climate season used for the reconstruction were purely based on the highest correlation obtained, and all other considerations and potential implications were somehow overlooked.

**6. Regarding physiological explanation. Extended between lines 25 of page 11 to to 21 page of 12.in the new version of the ms.**

**Anonymous Referee #2. 25 February, 2016.**
The manuscript presents climate reconstruction for the 1602-2012 period for previously less explored part of the Iberian Peninsula. The authors used a new standardization method based on the basal area. The applied detrending improved the quality of the reconstruction since the size-based standardization maximizes the common signal. The presented reconstruction fills the gap in climatic reconstructions in the area vastly affected by climate change where long term climatic records (and old trees) are very rare.

The first version of the ms contained numerous small errors. Some of them were already commented by the referee #1. The authors have responded to them. Their response is to my opinion adequate. The authors presented their response and submitted an improved version of the manuscript. The errors are now corrected, although they generally did not affect the message (result and conclusions) of the manuscript.

The chronology is to my opinion well-constructed and I agree with the response of the authors. The fact that they used different species has been adequately justified already in the first version of the ms. This is a common praxis if the chronologies match well - this is the case in this ms. Both versions of the ms contain small errors (mainly language and style) which should be corrected at the end of the interactive review process.

**S. Klesse; stefan.klesse@wsl.ch; 01 March, 2016**
While the authors have addressed the remarks of Reviewer #1 in a reasonable and adequate way, I see some methodological problems, mainly with the RCS (baspois) application:
1. Did you use pith offset (or for your case of Basal-Area-RC distance to pith) estimates? I cannot find it in the text. If not, why? Omitting pith offset estimates will lower your RC and ultimately introduce a fake negative trend in the early years of your chronology. In your case of inversion a positive biased trend, which could be amplified when using Basal-Area. See Briffa & Melvin 2011 _"A closer look on RCS..." and Klesse & Frank 2013 (attached).

**Dear Stefan:**
**1. Regarding pith offset. On the one hand, we use a set of sites (VIN, CAV, NEI, URB, COV, HER) of which the samples are available, and therefore pith offsets can be estimated. On the other hand, we use a set of sites from the ITRDB (s047, s048, s049, s050, s006) from which the samples cannot be accessed. Here, for each tree, we have assumed PO = 1 in the oldest sample, and adjusted the shorter series accordingly. Sure this procedure introduces uncertainties, but this is true for all studies using data from the ITRDB. We believe, however, that these uncertainties do not generate a systematic bias, but are minimized using the new BasPois detrended method based on basal area instead of age.**

2. You include old ITRDB datasets from the 1980s. Do you have the samples or pith offset estimates, or at least correct for the a- and b-sample difference of starting year? For example: a- and b-samples of the ITRDB series SPAI047 have quite large differences between their starting years (mean: 33 years). For their RCS curve that would mean, that those samples are overestimated on average already by 50mm (should be probably 0.5mm).

**2. Perhaps already addressed in the previous paragraph. Since we do not have the ITRDB samples, pith offsets need to be estimated. Assuming an age of 1 in the oldest sample of every tree (of the ITRDB data) is indeed a compromise, and perhaps there are other possible solutions, however, we believe that for the development of a chronology representative of the regional climate of the western Iberian Range, all available samples should be included.**

2b. The y axis in figure 4b is presumably off by a factor of 100 and should range from zero to 2.5mm instead of 250mm.

**2b. We apologize for the mistake. The axis is now correctly labelled (Fig.4).**

3. Do the trees have the same growth rates at all 11 sites? If not then a use of a single RC might introduce false trends, when sample and site replication changes. From originally 11 sites, 5 drop out in the 1980s, 4 in the 90s and you are left with only two sites. Do these sites have the same growth level as the ones that drop out before? See also Figure 6 in Klesse & Frank for an example of falsely introduced trends. I have attached a figure showing this potential problem including the 5 Iberian Range (IR) ITRDB chronologies and 2 chronologies from Büntgen near Madrid. Although 250km to SW they grew at similar elevation and correlate with the mean chronology of the other 5 series quite good (r=0.52, 1701-1985, 30-year spline detrended). So, well in the range of your observed site to site correlations and only a little bit weaker than your weakest site to regional chronology corrlelation, but completely independent (one could actually argue to include them to increase the regional representation, but that's beside the point here). I applied a single RC and no pith offsets, split the IR and Büntgen series and averaged them separately with an arithmetic mean. It is obvious that the mean of Büntgen have permanently lower values over the IR series. So if the IR series drop out, the overall RCS chronology gets heavily drawn towards lower values, while the Büntgen series remain constant. This effect could probably also be enlarged using the size/basal area detrending.
Can you show that this does not cause a problem in your data?

**3. About growth rates, the variability among sites is lower than the variability within a site. Besides, we are not joining chronologies, but develop the regional chronology from all 316 individual TRW series. It is generally unavoidable to add some noise when integrating TRW series from different locations and species in a regional chronology. However, the high correlation between each site and the regional chronology suggests a general climatic signal. Similar approaches have been detailed in Briffa et al. 1998 and numerous climatic reconstructions have been developed using networks of different sites and species i.e. Wilson et al. 2003, Battipaglia et al. 2010, Büntgen et al. 2011 or, Esper et al. 2012. However, to prove that the variable end dates are without effect in the trend we develop a regional chronology with the sites ending in 1993 (using the BasPois detrended method) (Fig1 on this comment).**

[Figure]

**Fig.1**

4. Figure 5a) Why do you compare your residual AC-free chronology with raw temperature data? That does not make sense if you want to highlight common signal in the high-frequencies. The simplest method would have been to detrend both series with a flexible spline (e.g. 30 years). That actually comes back to Remark 4 from Reviewer #1. If the TRW signal is truly representing pSep21 temperature, than it still should at least have reasonable negative correlations on the high-frequencies. A running correlation with raw temperature and BasPois does not answer Remark 4 and still contains trend-in-signal and not necessarily causal effect. I believe the authors might have kept things too easy during the RCS application, which might have led to erroneous conclusions. I would be really happy if my concerns don't have an impact on the conclusions, but without showing that I remain cautious.

**4. As suggested, we have detrended the climate data using a flexible spline (30 years) and correlated with ArstanRES and ArstanSTD chronologies emphasizing high frequency variance. The results show an increase in correlation with pSep21 temperature; for ArstanRES the correlation is r=-0.39, while for ArstanSTD the correlation increases to r=-0.56. These correlations indicate that the reconstruction contains some skill in the high frequency domain. Nevertheless, in order to assess long-term climate variability, we prefer using the BasPois chronology, in which the climate signal is enhanced, and both high and low frequency pSep21 variance and forcing (volcanic and solar) is retained. The changes in methodology and results have been included in the manuscript as well as the new Figure 5 (Fig2 on this comment).**

[Figure]

**Fig.2.**

**Anonymous Referee #1 interactive comment. 08 March, 2016**
The effort of the authors commenting and answering my questions and assuming mistakes in the calculation of some statistics is laudable. However, relevant issues described in detail in my initial review have not been answered and the reliability of the calibration is still questionable. The key component of this manuscript is the development of a climate reconstruction based on tree rings. If the calibration raises serious doubts, then the whole manuscript is dubious. I am aware of the effort that is needed to develop a proxy-based climate reconstructions, however, I still believe that the reconstruction presented in this manuscript is not fully reliable mainly due to three main issues:

**Referee #1: Thank you for your interest and comments, which we aim to answer here.**

1. How good or bad is the agreement between the tree-ring record and the climate record on the high-frequency domain remains unanswered. A poor visual agreement on the high frequency is obvious having a look to Figure 8, and the author's reply ("the reconstruction is better at mid and low frequencies than at high frequencies") is insufficient and does not provide any additional or new information to clarify this point. The way to solve the doubt is, as suggested in point 4 of my initial review and also suggested by S.Klesse, to do a comparison (correlation) of both series at different time scales and make sure that the correlation observed in the calibration are due also to synchrony in the high frequency and not only to a similar long-term trend. If the series correlate on the low frequency but do not show a reasonable agreement on the high frequency, then the correlation shown for the calibration period would be spurious and the reconstruction simply incorrect. The running correlation analysis will only answer this question if the series would have been high-pass filtered, which is not the case. In addition, the residual analysis now included on the paper would also require to include some test on the trend and autocorrelation of the residuals (i.e., Durbin-Watson test). If the calibration fulfils all above (agreement on the high-frequency domain and test of residuals), then we could talk about a statistical reliable calibration.

**1. In order to test the agreement between the tree-ring chronology and climate record in the high-frequency domain, and in line with Stefan Klesse's suggestions, we correlated the ArstanRES and ArstanSTD timeseries with the detrended (30-year spline) climatic data. The results show an increase in correlation with pSep21 temperature. For ArstanRES the correlation is r= -0.39, whereas for ArstanSTD the correlation increases to r=-0.56. These results validate correlation in the high-frequency domain and indicate that the reconstruction signal is not spurious. However, we intend to reconstruct both high and low frequency climate variations and prefer using the BasPois chronology as it enhances the climatic signal (r=-0.78) and reproduces the full variance spectrum of the pSep21 variable very well. We performed a Durbin-Watson test, as suggested, to assess residual autocorrelation. The results were added to Table2. The Durbin Watson value for the period 1945-2012 is 1.45 (p<0.001) indicating no substantial autocorrelation in the residuals.**
**We believe that both the correlations after removal of low frequency variance as well as the insignificant autocorrelation in the residuals support the pSep21 reconstruction.**

2. Whether chronologies encoding different climate signals have been merged into a final composite remains also unanswered. The new column added to Figure 3 containing the values of the correlations between the single and regional chronologies does not answer my initial question. Checking whether all chronologies encode the same climate signal means to correlate each individual chronology with climate. This is the way to know if tree growth is limited by the same climatic factor at all sites or different climate signals are being mixed in the regional chronology. Considering that the chronologies used are from different tree species, derived from different elevations and some chronologies do show poor correlation with the others, testing potential different climate signals is advisable, particularly because solving such a question is extremely easy.

**2. About chronology development, we did not merge site chronologies, but applied the standardization methods to all 316 individual TRW series to produce a regional chronology. Nonetheless, we added climate calibrations for each site to validate that the climate signal is regionally consistent. We developed a chronology for each of the 11 sites (detrended with the BasPois method) and correlated with the climatic variables. Highest correlations in the 11 sites appear for pSep20, pSep21 and pOct21. Since we chose to reconstruct pSep21 we also performed running correlations using a 30-year window to assess correlation stability within the calibration period. Results are shown in the Fig.1 (of this comment) and chronologies are sorted by elevation, VIN and CAV are *Pinus uncinata*, while the rest are *Pinus sylvestris*. The correlation never drops below r= -0.2. There are also periods surpassing r=-0.80. However, we would like to reemphasize that the aim of this study it is not to develop a local climate reconstruction, but to reconstruct the regional climate of the western Iberian Range.**

[Figure]

**Fig.1.**

3. The physiological explanation is still too general (in fact, has not substantially changed) and not very convincing. It is hard for me to picture how tree growth can be negatively influenced by the cumulative mean of temperature from the current and previous year of growth: how trees manage to grow then? How did they survive for centuries and did not die by carbon starvation if cumulative temperature of the previous 21 months have no positive effect on growth? Physiologically seems quite unlikely to me but still, I was hoping for a good explanation or answer that could challenge my thoughts on this regard.

**3. We would like to remark that tree-ring growth it is not negatively influenced by temperature. It is, however, negatively correlated with temperature of the previous year using a cumulative monthly mean of 21 months. That would mean that within the environment in which trees are growing and with respect to the mean, they will grow more in cold years than in hot years.**

**The negative temperature correlation is already shown for the previous September (r=-0.56) without any cumulative monthly mean. This negative temperature correlation has been reported in numerous dendroclimatic studies (i.e. Büntgen et al. 2006 or van der Werf et al. 2007) including the most recently developed climatic reconstruction for the Iberian Peninsula by Dorado-Liñán et al. 2014 showing a negative correlation with previous summer temperatures. One of the strengths of this paper is precisely adding the cumulative monthly mean to the climate variables which maximizes the correlation to r=-0.78.**

**The ecophysiological explanation of previous year's influence on current's year tree-ring growth was already related with the storage of starch and sugar in parenchyma ray tissue and the remobilization of carbohydrates from root structures. Memory effects on TRW data have also been studied regarding the delayed response in TRW to post volcanic eruptions (1´Lij5 years) associated with a decrease in current's year temperature (D'Arrigo et al., 2013, Esper et al., 2014).**

**We agree on the need to conduct further studies to better understand the full range of ecophysiological processes of pine and other species. To this extend, we are aware of an experiment conducted by a colleague (Dr. Eustaquio Gil Pelegrin; ttps://www.researchgate.net/profile/Eustaquio_Pelegrin) in which they try to demonstrate**

**that the generation of pinecones and needles in pine trees is very slow and itgenerally takes two years.**

C.Salt; christ.salt77@gmail.com: 09 May, 2016
After reading this discussion paper, I was left with the doubt on whether the authors have reconstructed temperature or precipitation. Considering that all or most of these sites are probably  sensitive to variations in soil moisture given their location in Mediterranean mountains, at least a mixed precipitation-temperature signal could be expected and should be analyzed and discussed. One must be extremely careful when analyzing negative effects of temperature on tree growth, particularly at sites >1500 m asl where temperatures are most likely not warm enough to cause direct damage to plant cells.
What would be the biological mechanism of a 21-month long cumulative negative effect of temperature, if it were not for an indirect effect through hydric stress of the trees? It makes sense that these relationships are driven by temperature increasing the evaporative demand or vapor pressure deficit. Thus, precipitation or a drought index should be considered in the analysis. I don't think this issue has been addressed in the original file or in the author'sresponse to referee #1. Results for SEA for volcanic eruptions would show lower temperature 3 years decrease after eruptions. That would mean wider tree rings. But those could also be caused by increased precipitation as shown for other parts of the Mediterranean Basin (Köse, N. et al., 2013. An improved reconstruction of May-June precipitation using tree-ring data from western Turkey and its links to volcanic eruptions. International Journal of Biometeorology, 57(5): 691-701.)

I would suggest the use of partial correlations for temperature (secondary variable), controlling for the effect of precipitation (primary variable). Using something like the seascorr function [Meko, D.M., Touchan, R. and Anchukaitis, K.J., 2011. Seascorr: A MATLAB program for identifying the seasonal climate signal in an annual tree-ring time series. Computers & Geosciences, 37(9): 1234-1241] should be straightforward. I would recommend the same time periods and lags be analyzed for precipitation or a drought index (similar to figure 5), before performing a temperature reconstruction from negative correlations with tree rings. It may be that the correlations with temperature are ok, but I think this deserves better explanations and justifications.

**Dear Chris, thank you for your interest and comments, which we aim to answer here.**

**Speaking about the Iberian Peninsula can sometimes generate misconceptions. The Iberian Peninsula is a very large territory with a broad set of climates ranging from a dry Mediterranean climate with 200 mm/year and a dry season during summer to an Atlantic climate with more than 2,500 mm/year and no dry season. The study area, as described in lines 23 to 27 (page 3) belongs to a Continental bioclimatic belt which is characterized by moderate mean temperatures (9.5C) and a mean annual precipitation which exceeds 1,000 mm/years very frequently (Fig 2A,Fig.2AC in the manuscript).**

**Therefore, there is no dry season within the study area. As well as in other mountain forests in Spain (see Büntgen et al., 2008, Dorado-Liñán et al., 2014), trees in the study area are limited by temperature. In Dorado-Liñán et al. 2014 they reconstruct the previous year's summer temperature for the past 800 years in the southeast of Spain using tree-ring width. During the conduct of the first analysis, we also took into account precipitation and drought indices such as SPI (McKee et al., 1993) and SPEI (Vicente-Serrano et al., 2010). However, due to the poor correlation values (see Fig.1 in the comment) we decided to focus on the maximum temperature signal. In Fig.1 of this comment SPEI (1 to 24) and SPI (1 to 24) values are correlated with the BasPois Chronology. The maximum correlation (r=0.35) is shown for**

**the SPEI19 of August and it is very much related with the temperature, since the SPEI drought index integrates temperature, in terms of evapotranspiration, to the equation.**

**There are, however, as suggested, some mountain areas in the Iberian Peninsula with Mediterranean climate conditions including a dry season with its trees limited by precipitation. For instance, in Tejedor et al., 2015 we developed a drought reconstruction using the SPI index.**

Büntgen, U., Frank, D., Grudd, H., Esper, J.: Long-term summer temperature variations in the Pyrenees. Climate Dynamics, 31 (6), pp. 615-631, 2008.

Dorado Liñán, I., Zorita, E., González-Rouco, J.F., Heinrich, I., Campello, F., Muntán, E., Andreu-Hayles, L., Gutiérrez, E.: Eight-hundred years of summer temperature variations in the southeast of the Iberian Peninsula reconstructed from tree rings. Climate Dynamics, 44 (1-2), pp. 75-93, 2014.

McKee TB, Doesken NJ, Kliest J (1993) The relationship of drought frequency and duration to time scales. In: Proceedings of the 8th Conference on Applied Climatology, Anaheim, CA, USA, 17–22. American Meteorological Society, Boston, MA, USA, pp 179–184

Vicente-Serrano SM, Beguería S, López-Moreno JI (2010) A multiscalar drought index sensitive to global warming: the standardized precipitation evapotranspiration index. J Clim 23(7):1696–1718.

Tejedor, E., de Luis, M., Cuadrat, J.M., Esper, J. & Saz, M.A. 2015. Tree-ring-based drought reconstruction in the Iberian Range (east of Spain) since 1694. International Journal of Biometeorology, DOI:10.1007/s00484- 015-1033-7.

[Figure]

[Figure]

Fig.1

**Anonymous Referee #3: 06 July, 2016.**
I find this paper intends to show and interesting matter, it is well written, and with a good quality of figures. Only some typos need to be corrected. However, there are some aspects that need clarification before the paper can be ready for publication, and some of them make me doubt about the validity of the results.

First of all, I wonder what the real objective of the paper is. On the one hand, different standardization methods are tested, and on the other hand the authors perform a reconstruction of a climatic variable. Though I can understand that this is a necessary step to provide a reliable reconstruction, I do not see that has this been brought in detail into the discussion, especially as regards the first two methods. But my main concern regards the variable selected for reconstruction. Though statistics seem to me optimal, I am not able to figure out what the causes for the existent relationship could be (21 month temperature). Explanations in the Discussion are too weak to be convincing, and I think this aspect needs to be much better clarified or hypothesized.

**Referee #3:** Thank you for your interest and comments. The issues raised in your comments have been deeply treated throughout the open discussion process. We believe that in the latest version of the manuscript questions related with the standardization method and the 21 months climate variable have been clarified. In any case, we will be glad to address any particular aspect if there is a further suggestion. Attached is the latest version of the manuscript.

[revised manuscript text omitted]

---

## Author Response (AR2)

**Dear Editor,**

**This is the Point-by-point response to Report#2 with which we aim to answer all the suggestions and comments the reviewer made.**

More important comments:

1. One of the advantages of the RCS standardization method is that it is usually presented as the method that best retains low frequency variability. This study presents another variant of RCS, based on an alignment of trees according to their size instead as according to their age. The obvious question would be how this variant behaves in the low-frequency domain compared to the RCS method. There is barely a word on this in the manuscript. As the bare minimum, both chronologies should be shown together in a figure, comparing the amplitude of low-frequency variations.

**More detailed discussion on the RCS and BasPois methods is addressed in lines 35-49 of page 7, and 1-13 of page 8. In addition, as suggested by the reviewer, the Fig.6 now includes the additional standardization methods.**

Related to this comment, it is unclear to me how the trees are really aligned in the BasPois chronology. The study mentions 'the 'square of the basal area' as the independent variable, but Figure 4 indicates cm2 es the units of the square of basal area. I think this is an error, and that more logically it is the basal area (not its square) the independent variable. This would match the units in Figure 4. Otherwise, I do not see the rationale of using the square of the area (it is probably the square of the diameter of the ring ?)

**We apologized for the misunderstanding and have correctly mentioned the 'basal area'.**

2. The physiological explanation for a response of the chronology to the mean of the monthly maximum temperature in the previous 21 months is basically a bit of hand waving. Maybe there is at this point no other explanation, but the authors could just candidly say it.

**We have delved deeper on the possible explanation of the 21 months in the discussion, lines 10-18 page 9.**

More particular points

3. The IPCC report, explains the acronym IPCC. All acronyms should be spelled out, even if they are obvious.

**As suggested by the reviewer we have spelled out the acronym.**

4. a high-resolution temperature reconstruction. High-resolution in time, I guess. The sentence in this context is unclear as it could refer to a spatially resolved reconstruction over this region

**As suggested by the reviewer we have rephrased the sentence.**

5. Iberian Range. Explain where the Iberian Range is located, at least broadly. The map shown in Figure 1 is of low-quality (at least in this pdf file) and should be improved for publication

**As suggested by the reviewer we have included a reference to Figure 1 to guide the readers. We apologized for the low quality of the figure, which is low due to compression to .pdf format. Final figures will have high quality.**

6. 'RE is a measure of shared variance between actual and estimated series ' This explanation of RE is not totally correct/specific. ( the correlation is also a measure of shared variance). RE is better defined a measure of the typical size of the errors relative to the typical size of variations relative to the calibration mean.

**As suggested by the reviewer we have improved the definition of RE. Lines 41 to 45, page 4.**

7. To transfer the TRW chronology into a temperature reconstruction a linear regression model was used. This is too unspecific. There are many variants of linear regression. I guess that the authors have set temperature as dependent variable and the chronology as independent variable, and have used Ordinary Least Squares assuming gaussian independent errors to estimate the regression coefficient. All this information is needed to completely specify the regression model

**As suggested by the reviewer we have improved the details of the linear regression used, lines 15-18 of page 5.**

8.with a gradual decline of the growth until the cambial of 450. Cambial age from 500 to 550 until the cambial age of 450 years, I guess. Cambial age from 500 to 6550 years (?)

**As suggested by the reviewer we have rephrased it to clarify.**

9. Calibration of the four differently detrended mean chronologies reveals a highly negative correlation with maximum temperatures. With maximum temperatures or with monthly mean of daily maximum temperatures ? Taken the sentence in the manuscript literally, the authors mean the maximum temperature attained within the 21-month window. I do not think this is what they really mean.

**As suggested by the reviewer we have correctly named the climate parameter as 'monthly mean of daily maximum temperature'.**

10. Correlations with previous-year September (r = -0.39), and the ArstanSTD chronology correlates at r = -0.56 with September and October temperature of the previous year with a cumulative monthly mean of 21 months. Here and everywhere in the manuscript, this way of specifying the calendar window with highest correlations is confusing. I agree that it is not straight forward to explain, but the characterization of September temperature is misleading. It is clearly not the September temperature, but the mean (maximum) temperature over a 21-month window. Thea authors should find a better way of defining this quantity. In this partucular sentence, I would suggest something like ' the 21-month mean temperature centered in September or October'. An efficient way would be to define it in the Methods section, attach an acronym to it (e.g. T_21_Sept or T_21-Oct or similar) and subsequently use the acronym throughout the manuscript.

**As suggested by the reviewer we have now used acronyms (explained in the methodology, lines 26-29, page 4) to indicate the climate parameter.**

11. 'and the first principal component explains about 35% of the variance'. Have the chronologies been standardized to unit variance before the PCA ? Otherwise the amount of explained variance is not informative, since it would depend on the individual variances of the chronologies

**Since PCA analyses is not used or commented anywhere else in the manuscript, we remove the sentence to avoid confusion.**

12. It is remarkable that the 12 years of the XXI century. In a scientific text, in English it is usual to refer to the 21st century, but this may be a matter of taste. Please, check.

**As suggested by the reviewer we now refer to 21$^{st}$**

13. ' The year-to-year temperature variability is ..' The reconstructions refer to a 21-month calendar window, so it is confusing to refer to the year-to year variability. This is actually not clearly resolved in the chronology. Perhaps better refer to high-frequency (biennial) variability

**As suggested by the reviewer we have rephrased to clarify, line 46 of page 6, lines 1 and 3 of page 7.**

14. The main driver of the large-scale character of the warm and cold episodes may be changes in the solar activity. This point can be quite controversial and it is definitively contrary to present understanding, which states that the most important forcing for the midlatitude temperature variations is the volcanic forcing. It also seems speculative in this text since the authors have not conducted any proper attribution test to separate solar and volcanic forcing, which are known to be strongly correlated.

**We agree that stratospheric sulfate aerosols from large volcanic eruptions are a prime forcing of past millennium climate variability. However, this forcing typically acts on shorter (inter-annual to decadal) timescales, and does not necessarily mean that all regional reconstructions are similarly affected by these events. A prominent continental scale temperature reconstruction for Europe (Luterbacher et al. 2016 in ERL) recently found better agreement with high-end estimates for total solar irradiance over the past millennium (also involving model simulations). So, while we generally agree with the reviewer, it still seems reasonable to report the correlations between solar forcing and our regional recon, and point the low degrees of freedom after smoothing the data.**

15. Overall, the correlation between the reconstruction and the solar activity is 0.34 (p < 0.0001), and increases to r = 0.49 after 11-year low pass filtering the series, thought the degrees of freedom are substantially reduced due to the increase autocorrelation. Another comment is that the number of degrees of freedom affects the statistical significance but not the magnitude of the correlation. A lower number of degrees of freedom does not per se on average artificially increase the correlation. thought - > though

**Here, we are not saying that we would expect a higher correlation after smoothing. What we say is just that the r =0. 49 correlation is based on less degrees of freedom due to the low-pass filtering. As suggested by the reviewer 'thought' has been corrected.**

16. The SEA (Fig.10) indicates some impact of volcanic eruptions on the short-term temperature variability within the reconstruction. It shows significance (p < 0.05) decrease in September's temperature with a lag of three years. The details of the SEA are obscure. The manuscript does not indicate which eruptions have been considered and how the significance has been established. This definitely needs a longer explanation

**The manuscript does indicate that the major volcanic eruptions that have been considered were those identified by Crowley (2000) line 12 of page 5. We agree with the reviewer that Crowley (2000) may seem slightly outdated dataset and yet, even though new volcanic reconstructions have been published such as Gao et al., 2008 or Sigl et al., 2015, over the past 400 years the events are quite well understood and well dated and hence we used the highly impact Crowley (2000) volcanic list. We have included the year of the volcanic events in lines 12 and 13 of page 5.**

**\*Note that the discussion has been reorganized.**

17. developed a 410-year maximum September temperature reconstruction developed a reconstruction of the monthly mean of daily maximum temperatures

**As suggested by the reviewer the sentence has been rephrased.**

18. signal to noise ratio, captures the regional climate signal accurately. A chronology never captures the climate signal accurately

**As suggested by the reviewer the sentence has been rephrased.**

19. In fact, climate variability is more size-dependent than age or species (De Luis et al., 2009). The impact of climate variability on trees may be more size-dependent, not the climate variability itself

**The discussion referred to this issue has now been improved from lines 42-48 of page 7.**

20. Memory effects in TRW data have been also studied regarding the delayed response in TRW (1~5 years) to post volcanic eruptions associated with a decrease in current's year temperature (D'Arrigo et al., 2013, Esper et al., 2014)....... According to the SEA (Fig.9), the volcanic eruptions have a significance reduction (95% confidence) of September's temperature (-1.98oC) with a three years lag.

This paragraph mixes two different effects, and it is not clear which one the authors are referring to. One effect is that the temperature response to eruptions is itself delayed, since volcanic aerosols need some time to spread globally in the stratosphere. The other effect is the physiologically delayed response of trees to sudden temperature drops ( or to reduction in sunlight caused by the eruption).

**The discussion has been here clarified in lines 20-39 of page 9.**

21. 2012 in agreement with the raise of temperatures observed for last decades rise in temperatures

**As suggested by the reviewer 'raise' was changed for 'rise'.**

22. between the chronology and the climate parameter chosen never drops from -0.54 below -0.54

**Changed as suggested by the reviewer.**

23. will trigger an incessant decrease in the tree-ring growth would also cause a continuous decrease in tree-ring growth

**Rephrased as suggested by the reviewer.**

24. Even though the CRU dataset extents the 1901-2013 period

The reader will wonder how is it possible that the CRU temperature records start in 1901 whereas the coverage of most meteorological stations starts in 1950. This would raise doubts on the quality of the pre-1950 temperature data

**Prior to 1950 just some areas of Spain have a good coverage of meteorological stations. In the study area (Iberian Range) local instrumental weather stations are in fact not available prior to 1945. Since the CRU dataset interpolates climatic data sometimes within distances of more than 100 km, we focused on the generalized instrumental period in Spain to avoid including more bias than benefits by extending the calibration/verification period.**

25. Even though the CRU dataset extents the 1901-2013 period, the general distribution of meteorological observatories in Spain did not begin until the mid-twentieth century (Gonzalez-Hidalgo et al. 2011)

spans the 1901-2013 period

**Modified as suggested.**

26. However, based on a TRW chronology, it is remarkable the high correlation coefficient for the full calibration period and the CRU dataset (r = -0.78).

the high correlation coefficient is remarkable

**Modified as suggested.**

27. However, previously to the Dalton minimum prior to the Dalton Minimum

**Modified as suggested.**

28. Overall, the correlation between the reconstruction and the solar activity is 0.34 (p < 0.0001), and increases to r = 0.49 after 11-year low pass filtering the series which reconstruction of solar activity is being used here ?

**This phrase has been removed from the discussion.**

29. Figure 4, caption square of the basal area ? or basal area?

**Basal area, we apologized for the mistake. Right citation has been corrected in the manuscript.**

30. Figure 5, Figure 7 caption mean of daily maximum temperature

**Changed as suggested.**

Figure 9 purple shading indicates the mean square error based on the calibration I cannot see any purple shading in this pdf file

**Due to .pdf compression some figures have lost image quality.**

Solar forcing: which reconstruction of solar activity has been used here ?

**The solar forcing published by Crowley (2000). Now indicated.**

32. Figure 10, how many eruptions and which eruptions have been used

**Indicated in lines 12 and 13 of page 5.**

[revised manuscript text omitted]